# MSpecTmol: A Multi-Modal Spectroscopic Learning Framework for Automated Molecular Structure Elucidation

**Wenjie Du**[1,2,*,†]**, Xiaohan Qin**[1,2,*]**, Ye Wei**[3]**, Jun Xia**[4]**, Yang Wang**[1,2,†]
[1]**University of Science and Technology of China, China**
[2]**Suzhou Institute for Advanced Research, USTC, China**
[3]**City University of Hong Kong**    [4]**Hong Kong University of Science and Technology**
{duwenjie, angyan}@ustc.edu.cn;  junxia@hkust-gz.edu.cn

Reviewed on OpenReview: https://openreview.net/forum?id=kRhf5Z1Cy1

## Abstract

Spectroscopic techniques are indispensable for the elucidation of molecular structures, particularly for novel molecules with unknown configurations. However, a fundamental limitation of any single spectroscopic modality is that it provides an inherently circumscribed and fragmented view, capturing only specific facets of the complete molecular structure, which is often insufficient for unequivocal and robust characterization. Consequently, the integration of data from multiple spectroscopic sources is imperative to overcome these intrinsic limitations and achieve a comprehensive and accurate structural characterization. In this work, we introduce **MSpecTmol**, a novel **M**ulti-modal **Spec**trum information fusion learning framework for automated **Mol**ecular structure elucidation. By extending information bottleneck theory, our framework provides a principled and adaptive approach to fusing spectra. It designates a primary modality to extract core molecular features while leveraging auxiliary inputs to enrich the representation. To validate the end-to-end effectiveness of our framework, we design a two-fold evaluation: molecular substructure classification to probe its discriminative power in identifying substructures, and extends this knowledge to reconstruct plausible 3D structures. Our results not only demonstrate state-of-the-art performance in molecular substructure classification but also achieve near-experimental accuracy (~0.68Å) in molecular conformation reconstruction. These findings underscore the model's capacity to learn interpretable features aligned with chemical intuition, thereby paving the way for future advances in automated and reliable spectroscopic analysis.

## 1 Introduction

The rapid advancements in artificial intelligence (AI) have significantly propelled research in the chemical sciences (Goh et al., 2017; Chen et al., 2026; Du et al.), enabling breakthroughs in molecular property prediction (Feinberg et al., 2018; Walters & Barzilay, 2020), drug design (Blundell, 1996; Riccardi et al., 2018), and drug-drug interaction studies (Zhao et al., 2024; Wang et al., 2024). AI not only achieves high-precision predictions without compromising accuracy but also enhances trust in its applications through interpretable models (Chander et al., 2024; Rane et al., 2024; Qin et al., 2026). These developments have increasingly integrated AI into chemistry as an indispensable tool. Notably, the vast majority of existing studies are *post-designed*, meaning that they operate on molecules with known structures, represented either as molecular graphs or SMILES strings (Du et al., 2023; Xia et al., 2023).

However, for a novel, unknown molecule, chemists must first determine its fundamental structure before exploring its properties (Hastings et al., 2021; Stanzione et al., 2021; Qin et al.). In such cases, spectroscopic

---

*: First author.
†: Corresponding author.

techniques serve as powerful tools for structural determination, fundamentally projecting high-dimensional chemical structures into lower-dimensional spectral representations as present in Figure 1 (Barone et al., 2021; Meza Ramirez et al., 2021). Spectral techniques such as nuclear magnetic resonance (NMR), infrared (IR) spectroscopy, and mass spectrometry (MS) could provide critical insights into molecular structures (Fontana & Widmalm, 2023; Manogaran et al., 2024), including the presence or absence of functional groups (Ge et al., 2021) which plays a crucial role in confirming structural assignments and ensuring the reliability of downstream chemical analysis.

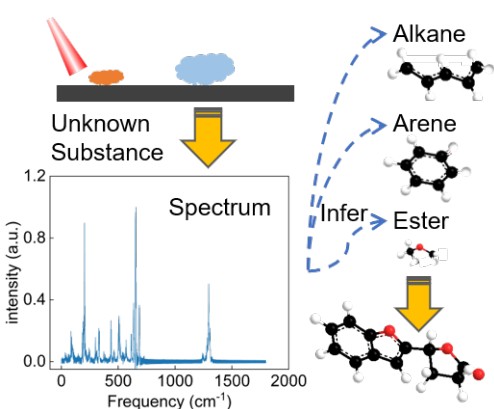

Figure 1: Schematic representation of inferring an unknown substance using spectral analysis techniques.

Yet, the inherent limitations of individual spectroscopic methods, due to their low-dimensional characteristics and the restricted information they contain (Xue et al., 2024; Bose et al., 2021), necessitate the integration of multiple spectroscopic sources to achieve more precise molecular determination. For instance, IR spectroscopy focuses on functional group vibrations, ultraviolet-visible (UV-Vis) spectroscopy reflects overall molecular properties, and NMR provides information about the local atomic environment (Chen et al., 2023; Manogaran et al., 2024). Each spectroscopic modality encapsulates distinct representational features and operates within different physically meaningful ranges (Barone et al., 2021). Therefore, a key challenge lies in fully leveraging the available spectroscopic data to extract its physical significance and enable accurate molecular structural determination (Meza Ramirez et al., 2021; Wang et al., 2026).

In this work, we propose a novel **M**ulti-modal **SpecT**rum information fusion learning framework based on information bottleneck theory for **Mol**ecule confirmation, termed **MSpecTmol**, to integrate multi-modal spectroscopic data. Our framework adopts a primary-auxiliary synergistic modeling approach, where the roles of primary and auxiliary representations are clearly delineated. By extending the multi-objective information bottleneck theory to this setting, we enable the primary modality to capture core information by filtering out redundant or irrelevant features, while the auxiliary modalities supplement the primary representation to enhance and refine the results. To comprehensively validate our framework's end-to-end effectiveness across the entire spectrum-to-molecule workflow, we rigorously applied it to two critically important tasks: molecular identification and spectrum-conditioned molecular conformation generation. In the molecular identification task, MSpecTmol significantly outperformed state-of-the-art baseline methods across both simulated and experimental spectra, achieving an F1-score of 0.959. Furthermore, the framework similarly demonstrated its capabilities for intricate structural elucidation in the challenging spectrum-conditioned conformation generation task, achieving an average RMSD of 0.682Å. Moreover, MSpecTmol captures critical spectroscopic fragments that align well with chemical intuition, providing a degree of interpretability for its predictions. The synergy between primary and auxiliary modalities offers a flexible strategy for researchers to adapt to various chemical challenges, further improving performance outcomes. We envision that molecular identification through spectroscopic data will become a key research focus in automated laboratory workflows. MSpecTmol represents a promising solution to this challenge, offering a robust and interpretable framework for this.

## 2 Related work

### 2.1 Spectroscopy-Based Molecular Modeling

Machine learning has advanced spectroscopy-based molecular structure prediction significantly. MS2SMILES (Liu et al., 2023) treats hydrogen atoms as implicitly linked to heavy atoms, improving molecular generation accuracy. DeepEI (Ji et al., 2020) serves as a deep learning framework for elucidating structures from EI-MS spectra. NEIMS (Wei et al., 2019) is a neural network model that captures fragmentation patterns from electron ionization for rapid small molecule mass spectrum prediction. Additionally,

ZODIAC (Ludwig et al., 2020) leverages tandem mass spectrometry (MS/MS) for molecular formula generation.

More recently, the rapid development of generative AI has expanded spectral analysis into the realm of Large Language Models and Diffusion Models. For instance, DiffSpectra (Wang et al., 2025) introduces a generative framework that formulates structure elucidation as a conditional diffusion process, enabling the end-to-end generation of 3D molecular conformations from multi-modal spectra. In parallel, Large Language Models have been adapted for this domain: SpectraLLM (Su et al., 2025) and MolSpectLLM (Shen et al., 2025) leverage the reasoning capabilities of heavy transformer backbones to treat spectrum-to-structure translation as a sequence generation task, bridging spectroscopy with textual molecular representations. Furthermore, SpectrumWorld (Yang et al., 2025) expands this frontier by introducing a multi-modal agent framework and benchmark suite to systematize deep learning research in spectroscopy.

While these methods have achieved notable success, they rely heavily on mass spectrometry, overlooking multi-type spectroscopic integration. Additionally, mass spectrometry is costly, sensitive to interference, and challenging to standardize in automated workflows. Recently, Alberts et al. released a 794k Multimodal Spectroscopic Dataset (Alberts et al., 2024), providing a foundation for integrating multi-spectroscopic data. Their work introduced baseline models for tasks like molecular structure prediction and functional group identification from spectral data, forming a key resource for our research. These studies highlight the potential and limitations of current methods, motivating our approach to integrate multi-spectroscopic modalities for enhanced molecular structure determination.

### 2.2 Information Bottleneck (IB) Theory

The IB theory provides a principled framework for extracting compact and informative substructures from complex data, playing a key role in challenges like denoising and compression. PGIB (Yu et al., 2020) extends IB by introducing a framework with a mutual information estimator for irregular graph data, and a connectivity loss to stabilize information extraction. VGIB (Yu et al., 2022a) further improves stability by injecting Gaussian noise into node representations, regulating information flow between original and perturbed graphs. Lee et al. (Lee et al., 2023) expanded IB to paired graphs with the Conditional GIB, optimizing compressed information extraction by retaining only the most relevant information. While effective, these approaches focus on single-target tasks and lack strategies for redundancy reduction and complementary integration under multi-modal conditions. This growing body of work underscores the versatility of IB theory while highlighting opportunities for further refinement, particularly in handling multi-modal scenarios, where redundancy removal and cross-modal synergy are essential.

## 3 Primary-Auxiliary Information Bottleneck (PA-IB)

In this work, we focus on learning the core representations $T_m$ and $T_a$ from the input primary spectrum $X_m$ and auxiliary spectra $X_a$.

**Primary-Auxiliary Information Bottleneck (PA-IB).** Given the primary spectrum $X_m$, auxiliary spectra $X_a$, and target $Y$, PA-IB compresses $X_m$ into a bottleneck $T_m$ while preserving information needed to predict $Y$, and compresses $X_a$ into a bottleneck $T_a$ while preserving the information needed to predict $Y$ conditioned on $T_m$. Formally, this corresponds to the constrained optimization:

$$\max_{T_m, T_a} \; I(Y; T_m) + I\big(Y; T_a \mid T_m\big) \quad \text{s.t.} \quad I\big(X_m; T_m\big) \leq I_c^m, \quad I\big(T_a; X_m, X_a\big) \leq I_c^a, \tag{1}$$

where $I_c^m$ and $I_c^a$ denote the maximum information the primary and auxiliary bottlenecks are permitted to retain. Introducing non-negative Lagrange multipliers $\alpha$ and $\beta$ yields the equivalent unconstrained objective:

$$\min -I(Y; T_m) - I\big(Y; T_a \mid T_m\big) + \alpha \, I\big(X_m; T_m\big) + \beta \, I\big(T_a; X_m, X_a\big), \tag{2}$$

where $\alpha$ and $\beta$ are Lagrange multipliers that balance the mutual information terms.

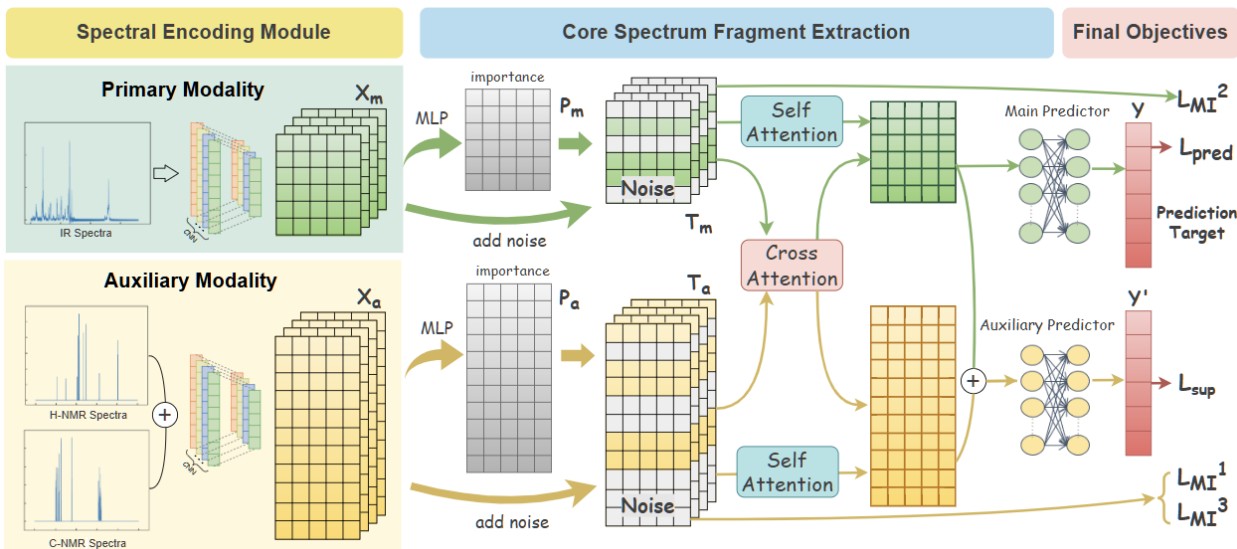

Figure 2: Illustration of the **MSpecTmol** framework. The model processes one primary and multiple auxiliary spectral modalities. Initially, vector representations are generated via linear interpolation and then fed into 1D-CNN layers to extract feature matrices. Subsequently, the core IB-Spectra module distills essential information from the primary modality and complementary features from the auxiliary inputs, producing a compact feature vector for downstream classification.

*IB-Spectra.* Given a set of spectra $(X_m, X_a)$ and its corresponding label information $\mathbf{Y}$, we identify the optimal primary spectrum $\mathcal{T}_{MIB}$ and auxiliary spectrum $\mathcal{T}_{AIB}$ under the PA-IB principle:

$$\mathcal{T}_{\text{MIB}}, \mathcal{T}_{\text{AIB}} = \underset{\mathcal{T}_{\text{MIB}}, \mathcal{T}_{\text{AIB}}}{\arg\min} \left[ -I(Y; T_m) - I(Y; T_a \mid T_m) + \alpha\, I(X_m; T_m) + \beta\, I(T_a; X_m, X_a) \right], \tag{3}$$

This objective involves the following four components:

- $-I(Y; T_m)$: Encourages the primary representation $T_m$ to preserve the most predictive information about $Y$. This term corresponds to the classical IB objective and ensures that $T_m$ serves as the main carrier of task-relevant information.
- $-I(Y; T_a \mid T_m)$: Drives the auxiliary representation $T_a$ to complement $T_m$ by capturing additional information that is not contained in $T_m$, thus improving the overall predictive capacity.
- $\alpha\, I(X_m; T_m)$: Regularizes the complexity of $T_m$ by penalizing excessive mutual information with the input $X_m$, thereby promoting a compact and generalized encoding.
- $\beta\, I(T_a; X_m, X_a)$: Limits the complexity of $T_a$ by minimizing its mutual information with the combined inputs $(X_m, X_a)$, encouraging selective representation of auxiliary information.

Combining these four terms, the final objective is: The primary representation $T_m$ captures the core information necessary for predicting $Y$. The auxiliary representation $T_a$ complements $T_m$ by providing any additional information needed for $Y$, while maintaining low complexity.

## 4 Methodology

### 4.1 Spectral Encoding Module

Here, $\{X_m, X_a\}$ denote a pair of input spectra, with $X_m$ as the *primary* spectrum and $X_a$ as the *auxiliary* spectra. To unify dimensions, each spectrum is interpolated to 600 uniformly spaced points by linear

interpolation, where the point value $x$ is updated to $X'$:

$$X'(x) = X(x_i) + \frac{(x - x_i)}{(x_{i+1} - x_i)} \cdot (X(x_{i+1}) - X(x_i)), \quad x \in [x_i, x_{i+1}] \tag{4}$$

where $x_i$ and $x_{i+1}$ are consecutive points in the original spectrum, and $X(x)$ represents the updated value at $x$. This interpolation step preserves the continuity of the spectral signal and ensures that each index corresponds to a consistent physical frequency, maintaining the integrity of peak positions, shapes, and relative intensities across all samples. This procedure is applied independently to two different spectra designated as auxiliary inputs. After interpolation, both spectra are normalized and concatenated to create a single 1200-dimensional auxiliary spectrum, $X_a$.

The input spectra $X_m$ and $X_a$ are encoded through two stages of 1D convolutional layers, each followed by batch normalization, ReLU activation, and max pooling. The use of 1D convolutions is motivated by the sequential nature of spectral data, where capturing local patterns is essential for extracting meaningful features. Convolutional layers enable the model to automatically learn hierarchical representations and spatially invariant characteristics from the spectra.

$$O^m = \texttt{MaxPool1D}(\text{ReLU}(\text{BatchNorm}(\text{Conv1D}(X_m)))), \tag{5}$$

$$O^a = \texttt{MaxPool1D}(\text{ReLU}(\text{BatchNorm}(\text{Conv1D}(X_a)))), \tag{6}$$

Here, $O^m \in \mathbb{R}^{c \times d}$ and $O^a \in \mathbb{R}^{c \times d}$, where $c$ denotes the number of convolutional channels and $d$ represents the length after linear interpolation. The pool size is set to 2.

## 4.2 Core Spectrum Extraction

In this section, we extract core spectral segments by first transposing the frequency representations: $H^m = O_m^\top$, $H^a = O_a^\top$, where $\top$ denotes the transpose operation. For the primary spectrum, we compress $X_m$ into $T_m$ by injecting noise into its learned embedding, encouraging the model to suppress less informative frequency bands. For the auxiliary spectrum, we similarly derive $T_a$ from $X_m$, $X_a$, and $T_m$, guided by Equation 8. The key idea is to enable the model to inject noise into insignificant frequency bands while injecting less noise into more informative ones (Yu et al., 2022a). we could calculate the probability $p_i^m$ and $p_i^a$ using an MLP, i.e.,

$$p_i^m = \sigma\big(\text{MLP}(\mathbf{H}_i^m)\big) \quad p_i^a = \sigma\big(\text{MLP}(\mathbf{H}_i^m \parallel \mathbf{H}_i^a \parallel \mathbf{T}_i^m)\big). \tag{7}$$

where $\sigma(\cdot)$ denotes the sigmoid functio. With the $p_i^m$ and $p_i^a$, we replace $\mathbf{H}_i^m$ and $\mathbf{H}_i^a$ of frequency band $i$ with noise $\epsilon$, i.e.,

$$\mathbf{T}_i^m = \lambda_i^m \mathbf{H}_i^m + (1 - \lambda_i^m)\epsilon^m, \quad \mathbf{T}_i^a = \lambda_i^a \mathbf{H}_i^a + (1 - \lambda_i^a)\epsilon^a, \tag{8}$$

where $\lambda_i^m \sim \text{Bernoulli}(p_i^m)$ and $\epsilon^m \sim \mathcal{N}(\mu_m, \sigma_m^2)$. Here, $\mu_m$ and $\sigma_m^2$ are mean and variance of $\mathbf{H}^m$, respectively. Thus, the information of $X_m$ is compressed into $T_m$ with the probability $p_i^m$, by replacing unimportant frequency bands with noise. Similarly, for the core auxiliary spectrum, The information from $X_m$, $T_m$, and $X_a$ is compressed into $T_a$ with the same probability $p_i^a$.

Moreover, to make the sampling process differentiable, the Gumbel-Softmax is adopted (Maddison et al., 2016; Jang et al., 2016) for the discrete random variable $\lambda_i$, i.e.,

$$\lambda_i = \sigma \left( \frac{1}{t} \log \left( \frac{p_i}{1 - p_i} \right) + \log \left( \frac{u}{1 - u} \right) \right), \tag{9}$$

where $\sigma(\cdot)$ denotes the sigmoid function, $u \sim \text{Uniform}(0, 1)$, and $t$ is the temperature hyperparameter that is set to 1.0 in this work. A detailed sensitivity analysis of $t$ is provided in Appendix H.

## 4.3 Model Optimization

To train the model while simultaneously detecting the core primary and auxiliary spectra, we minimize the objective of Eq. 2. In the following sections, we provide the variational upper bound of each term, which is minimized during training.

### 4.3.1 Minimizing $-I(Y;T_m)$

**Proposition 3.1 (Upper bound of $-I(Y;T_m)$)** Given the primary spectra $X_m$, and its label information $\mathbf{Y}$, we have:

$$
\begin{aligned}
-I(\mathbf{Y};T_m) &\leq \mathbb{E}_{T_m,\mathbf{Y}}[-\log p_\theta(\mathbf{Y}|T_m)] \\
&= \mathbb{E}_{(\mathbf{Y},T_m)}\log\left[P_\theta\left(\mathbf{Y}\mid T_m\right)\right] + H(\mathbf{Y}) := \mathcal{L}_{\text{pred}},
\end{aligned}
\tag{10}
$$

where $H(\mathbf{Y})$ is the entropy of the label $\mathbf{Y}$, which is constant across the dataset and can be omitted in the optimization. $p_\theta(\mathbf{Y}|T_m)$ is the variational approximation of the true posterior $p(\mathbf{Y}|T_m)$. Minimizing this upper bound corresponds to minimizing the prediction loss $\mathcal{L}_{\text{pred}}(\mathbf{Y},T_m)$, which is modeled as the cross-entropy loss for classification. The proof can be found in Appendix C.1. This bound follows the standard variational formulation widely used in the IB literature, such as VIB Alemi et al. (2016).

### 4.3.2 Minimizing $-I\big(Y;T_a \mid T_m\big)$

**Proposition 3.2 (Upper bound of $-I\big(Y;T_a \mid T_m\big)$)** We decompose the term using the chain rule of mutual information:

$$
\begin{aligned}
-I\left(Y;T_a \mid T_m\right) &= -I(Y;T_a,T_m) + I(T_a;T_m) \\
&\leq \mathbb{E}_{(\mathbf{Y},T_a,T_m)}\log\left[P_\theta\left(\mathbf{Y}\mid T_a,T_m\right)\right] + \mathbb{E}_{t_m\sim p(t_m)}\left[\text{KL}\left(p(t_a\mid t_m)\|q(t_a)\right)\right] \\
&:= \mathcal{L}_{\text{sup}} + \mathcal{L}_{\text{MI}^1}.
\end{aligned}
\tag{11}
$$

Here, $\mathcal{L}_{\text{sup}}$ represents the supervised prediction loss $\mathcal{L}_{\text{pred}}(\mathbf{Y},T_m,T_a)$, which is implemented as cross-entropy for prediction. The second term, $\mathcal{L}_{\text{MI}^1}$, corresponds to the KL divergence between the posterior $p(t_a\mid t_m)$ and a prior $q(t_a)$, regularizing the relationship between the auxiliary spectra $T_a$ and primary spectra $T_m$. This divergence is minimized using variational inference, and is estimated by averaging over samples of $t_m$. Detailed derivations can be found in Appendix C.1.1. For tractability, both the posteriors $p(t_m \mid x_m), p(t_a \mid \cdot)$ and the priors $q(t_m), q(t_a)$ are assumed to be unimodal Gaussians, which yields the closed-form KL in Appendix C.1.2. Specifically, as shown in Appendix J, we investigate the impact of different prior distributions of $q(t_m)$ and $q(t_a)$ on model performance, and select the best prior distribution as the distribution for MSpecTmol. Although the chain rule of mutual information and the variational KL bound used above are themselves classical tools, their composition in service of an asymmetric primary–auxiliary IB is, to our knowledge, new. In contrast to prior multi-view IB methods such as VGIB Yu et al. (2022b) and CGIB Lee et al. (2023) that apply identical bounds across all views, Proposition 3.2 explicitly captures the incremental predictive contribution of the auxiliary modality beyond what the primary modality already provides, forming the technical foundation of PA-IB.

### 4.3.3 Minimizing $I\left(X_m;T_m\right)$

**Proposition 3.3 (Upper bound of $I\left(X_m;T_m\right)$)** We apply the variational approximation to bound the mutual information term:

$$
I\left(X_m;T_m\right) \leq \mathbb{E}_{t_m\sim p(t_m)}\left[\text{KL}\left(p(t_m\mid x_m)\|q(t_m)\right)\right] := \mathcal{L}_{\text{MI}^2}.
\tag{12}
$$

Here, $\mathcal{L}_{\text{MI}^2}$ corresponds to the KL divergence between the posterior $p(t_m\mid x_m)$ and a prior $q(t_m)$. The KL divergence is computed using variational inference and is estimated by averaging over samples of $x_m$. The detailed derivation is provided in Appendix C.1.2.

### 4.3.4 Minimizing $I\big(T_a;X_m,X_a\big)$

**Proposition 3.4 (Upper bound of $I\big(T_a;X_m,X_a\big)$)** We minimize the mutual information between the auxiliary spectra $T_a$ and both the primary spectra $X_m$ as well as the auxiliary spectra $X_a$:

$$
I\big(T_a;X_m,X_a\big) \leq \mathbb{E}_{t_a,x_a\sim p(x_m,x_a)}\left[\text{KL}\left(p(t_a\mid x_m,x_a)\|q(t_a)\right)\right] := \mathcal{L}_{\text{MI}^3}.
\tag{13}
$$

Here, $\mathcal{L}_{\text{MI}^3}$ represents the KL divergence between the posterior $p(t_a\mid x_m,x_a)$ and a prior $q(t_a)$. The KL divergence is estimated using variational inference, with derivations detailed in Appendix C.1.3.

### 4.4 Final Objectives

The final objective function used for training is given by:

$$\mathcal{L}_{\text{total}} = \mathcal{L}_{\text{sup}} + \mathcal{L}_{\text{pred}} + \mathcal{L}_{\text{MI}^1} + \alpha\,\mathcal{L}_{\text{MI}^2} + \beta\,\mathcal{L}_{\text{MI}^3} \tag{14}$$

where $\alpha$ and $\beta$ control the trade-off between prediction accuracy and compression. The detailed derivations and proofs for $\mathcal{L}_{\text{pred}}, \mathcal{L}_{\text{sup}}, \mathcal{L}_{\text{MI}^1}, \mathcal{L}_{\text{MI}^2}$, and $\mathcal{L}_{\text{MI}^3}$ are provided in above.

## 5 Experiment and Analyses

### 5.1 Datasets and setups

**Datasets.** We utilize the large-scale dataset from Alberts et al. (2024) for molecular structure elucidation, which contains 794K molecules with simulated IR, $^1$H-NMR, $^{13}$C-NMR, and MS/MS spectra. For 3D molecular conformation generation, we employ the QM9S dataset (Zou et al., 2023), providing 130K molecules with UV, IR, and Raman spectra paired with their ground-truth 3D conformations. To examine model performance under experimental conditions, we collect about 12K molecules from the National Institute of Advanced Science and Technology, SDBS Web (https://sdbs.db.aist.go.jp) with MS, $^{13}$C-NMR, and 1H-NMR spectra, since no multi-modal dataset with experimental spectra is currently available. Further details are in Appendix D.

**Baselines.** For the molecular classification task, we compare our model against four baselines. 1D-CNN Jung et al. (2023) applies stacked 1D convolutions directly to the spectral sequence and serves as a lightweight convolutional reference. Transformer Klein et al. (2018) encodes the spectrum as a token sequence with self-attention. Alberts et al. Alberts et al. (2025) is a transformer-based spectrum-to-structure model that sets recent benchmarks on IR elucidation, and Wu et al. Wu et al. (2025) employs patch-based self-attention over IR spectra for molecular-structure prediction. For conformation generation, as far as we known, no prior work has been published. We thus constructed two baselines. The first is to replace our spectral encoder with a standard attention module, while keeping the diffusion architecture and hyperparameters identical. The second approach adapts RDKit (Landrum et al., 2025), OpenBabel (O'Boyle et al., 2011), ConfGF (Shi et al., 2021), and GeoDiff (Xu et al., 2022). In particular, to incorporate spectral guidance into the conformation generation process, we enhanced GeoDiff—generating five candidate conformations from SMILES and selecting the one that best matches the input spectrum using a contrastive selector—a feature not available in the other models.

**Metrics.** For the substructure classification task, we use micro F1-scores and molecular prediction accuracy, where the latter is defined at the sample level. For the conformation generation task, performance is evaluated by the root mean square deviation (RMSD). All experiments are independently repeated across 8 different random seeds, with the dataset randomly split into 8:1:1 for training, validation, and testing in each run, and the average results with variances are reported. We provide detailed hyperparameter settings in Appendix A and a full complexity analysis in Appendix E.

### 5.2 Results on Functional Group Identification

MSpecTmol exhibits strong capability in identifying molecular functional groups and significantly outperforms baseline models, as shown in Table 1. In the single-modal setting, the performance of MSpecTmol is higher than transformer-based models, which demonstrates the effectiveness of our proposed architecture in capturing both local and global spectral features. However, when auxiliary spectra are introduced in the multi-modal setting, performance improves consistently across all models. Notably, MSpecTmol benefits the most, indicating its ability to effectively leverage additional information from auxiliary spectra to enhance substructure recognition and prediction accuracy. To investigate the optimal number of modalities and select the primary modality, Appendix E analyzes the computational overhead of MSpecTmol across varying input configurations, specifically focusing on execution time and memory consumption. Guided by the necessary trade-off between efficiency and performance, we identified a synergistic combination of three modalities: IR, $^1$H-NMR, and $^{13}$C-NMR as the ideal setup. This configuration delivers robust predictive

Table 1: F1-scores for predicting functional groups. For multi-modal settings, the primary modality is indicated in **bold**. Baseline models are invariant to the choice of primary modality, whereas MSpecTmol leverages this information to achieve superior performance. The best results are highlighted in **bold**, and the second-best are underlined. Entries are reported as mean$_{\text{(std)}}$ over 8 runs.

| Spectrum Config. | 1D-CNN | Transformer | Wu et al. | Alberts et al. | MSpecTmol |
|---|---|---|---|---|---|
| **Alberts et al. (Simulated Spectra)** | | | | | |
| IR | $0.895_{(0.002)}$ | $0.881_{(0.021)}$ | $0.886_{(0.013)}$ | $0.891_{(0.007)}$ | $\mathbf{0.923}_{(0.004)}$ |
| $^{13}$C-NMR | $0.674_{(0.056)}$ | $0.913_{(0.017)}$ | $0.914_{(0.004)}$ | $\underline{0.919}_{(0.012)}$ | $\mathbf{0.920}_{(0.013)}$ |
| $^{1}$H-NMR | $0.839_{(0.005)}$ | $0.935_{(0.031)}$ | $\underline{0.943}_{(0.036)}$ | $\mathbf{0.946}_{(0.027)}$ | $0.927_{(0.013)}$ |
| **IR** + ($^{13}$C-NMR, $^{1}$H-NMR) | $0.900_{(0.004)}$ | $0.936_{(0.013)}$ | $0.944_{(0.012)}$ | $\underline{0.947}_{(0.014)}$ | $\mathbf{0.959}_{(0.022)}$ |
| $^{13}$**C-NMR** + (IR, $^{1}$H-NMR) | $0.900_{(0.004)}$ | $0.936_{(0.013)}$ | $0.944_{(0.012)}$ | $\underline{0.947}_{(0.014)}$ | $\mathbf{0.957}_{(0.014)}$ |
| $^{1}$**H-NMR** + (IR, $^{13}$C-NMR) | $0.900_{(0.004)}$ | $0.936_{(0.013)}$ | $0.944_{(0.012)}$ | $\underline{0.947}_{(0.014)}$ | $\mathbf{0.956}_{(0.031)}$ |
| **IR** + (MS/MS$_{\text{pos}}$, MS/MS$_{\text{neg}}$) | $0.887_{(0.008)}$ | $0.911_{(0.003)}$ | $0.924_{(0.012)}$ | $\underline{0.931}_{(0.031)}$ | $\mathbf{0.944}_{(0.015)}$ |
| **SDBS Database (Experimental Spectra)** | | | | | |
| MS | $0.801_{(0.018)}$ | $0.826_{(0.021)}$ | $\underline{0.837}_{(0.015)}$ | $0.836_{(0.010)}$ | $\mathbf{0.847}_{(0.012)}$ |
| $^{13}$C-NMR | $0.729_{(0.033)}$ | $0.821_{(0.020)}$ | $0.833_{(0.014)}$ | $\underline{0.836}_{(0.011)}$ | $\mathbf{0.842}_{(0.015)}$ |
| $^{1}$H-NMR | $0.701_{(0.027)}$ | $0.779_{(0.025)}$ | $\underline{0.801}_{(0.019)}$ | $\mathbf{0.803}_{(0.018)}$ | $0.792_{(0.014)}$ |
| **MS** + ($^{13}$C-NMR, $^{1}$H-NMR) | $0.847_{(0.022)}$ | $0.858_{(0.019)}$ | $0.872_{(0.020)}$ | $\underline{0.881}_{(0.017)}$ | $\mathbf{0.913}_{(0.021)}$ |
| $^{13}$**C-NMR** + (MS, $^{1}$H-NMR) | $0.847_{(0.022)}$ | $0.858_{(0.019)}$ | $0.872_{(0.020)}$ | $\underline{0.881}_{(0.017)}$ | $\mathbf{0.894}_{(0.016)}$ |
| $^{1}$**H-NMR** + (MS, $^{13}$C-NMR) | $0.847_{(0.022)}$ | $0.858_{(0.019)}$ | $0.872_{(0.020)}$ | $\underline{0.881}_{(0.017)}$ | $\mathbf{0.909}_{(0.018)}$ |

power while maintaining manageable computational complexity. Additionally, we compared our method with recent LLM-based baselines, and the results are detailed in Appendix G.

In the process of functional group classification, as shown in Figure 3(b) and (c), we observe that the prediction accuracy tends to decrease as the number of functional groups and heavy atoms in a molecule increases. This is likely because greater structural complexity leads to more intricate and overlapping spectral signals, making it challenging to disentangle the features corresponding to individual substructures. Despite these challenges, our model consistently achieves superior performance relative to baselines, particularly for molecules with complex structures. This advantage is primarily attributed to the PA-IB framework, which selectively filters out redundant or non-informative signals and retains only the most relevant structural information, thereby enhancing predictive reliability. To further evaluate the model's real-world robustness in handling complex and unknown molecular configurations, we conducted a stress test on the top 10% largest molecules, as their dense and overlapping spectral signals represent a significant distribution shift. Detailed quantitative results and further analysis of this experiment are provided in Appendix I.

As shown in Figure 3(a), MSpecTmol achieves the highest macro-F1 score, indicating balanced performance across both common and rare substructures. In classifying 37 functional groups (Figure 3(g)), prediction accuracy varies due to intrinsic spectral differences. Groups with weak or overlapping signals, such as alkyl (-$CH_3$) and ether (-O-), are more challenging than those with distinct peaks like carbonyl (C=O) and hydroxyl (-OH). Nevertheless, MSpecTmol consistently outperforms baselines, benefiting from the PA-IB framework that filters redundant signals and preserves the most informative structural cues.

To evaluate the effectiveness of MSpecTmol on real-world data, we constructed a dataset of approximately 12K samples collected from the SDBS Web. As shown in Table 1, all models suffer from a noticeable performance drop under single-modality settings, which can be attributed to the limited amount of data and the inherent complexity of experimental spectra. However, in multi-modal settings, MSpecTmol achieves a substantial improvement over the baselines, reaching an F1 score of 0.913. This result highlights the practical value of MSpecTmol. Furthermore, given the data-hungry nature of CNN-based models and the

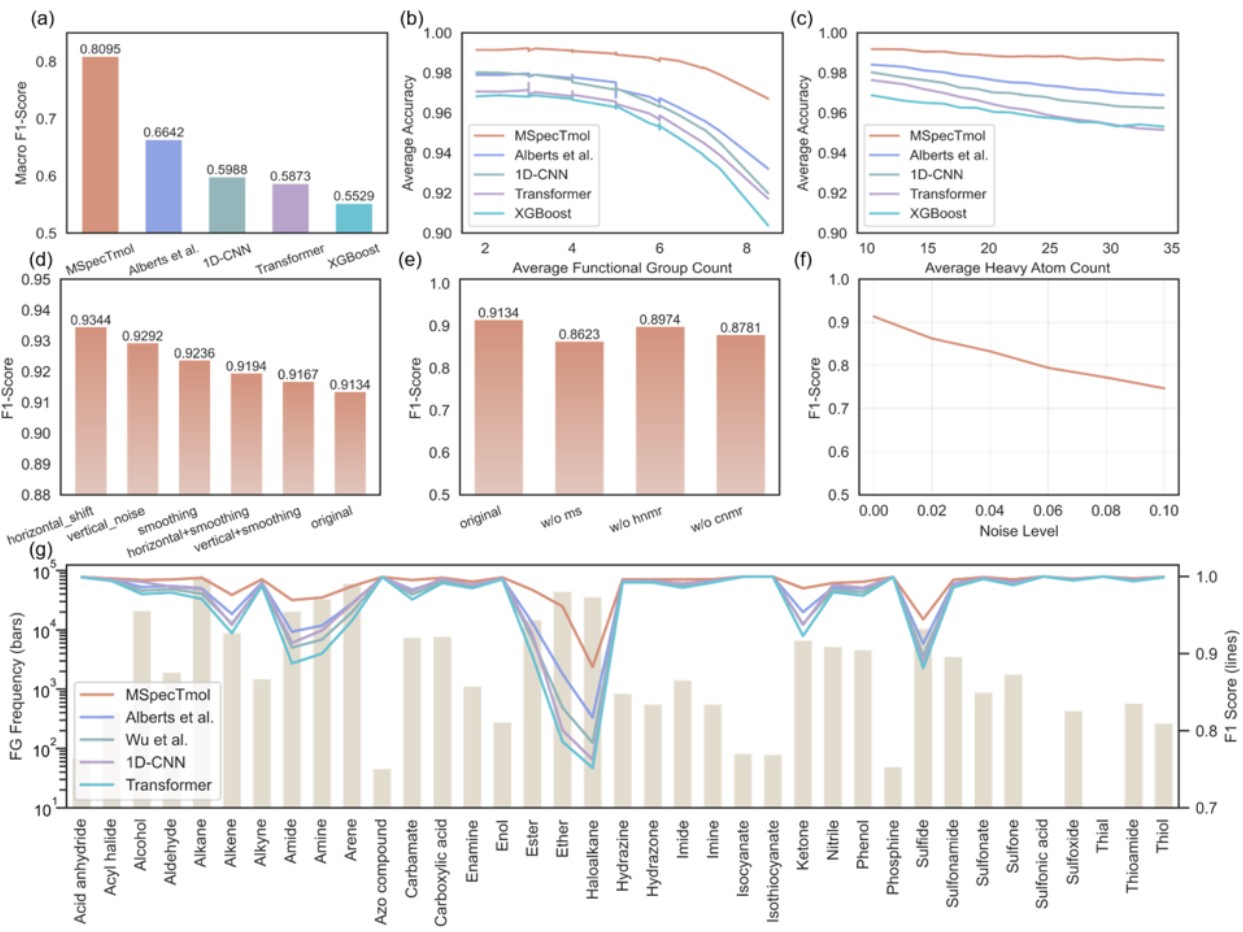

Figure 3: Comprehensive evaluation of MSpecTmol on functional group identification and robustness analysis. (a) Macro-F1 score results; (b) Performance across different numbers of functional groups; (c) Performance under varying heavy atom counts; (d) Analysis of data augmentation strategies. (e) Robustness test under missing spectral modalities (w/o MS, w/o $^1$H-NMR, w/o $^{13}$C-NMR; "w/o" denotes "without"). (f) Performance curve under increasing levels of Gaussian noise. (g) Correlation between the number of functional groups and prediction performance. Bars (left $y$-axis) show functional-group frequency; lines (right $y$-axis) show per-group F1-score.

scarcity of multi-modal experimental datasets, we investigated various data augmentation strategies. As illustrated in Figure 3(d), horizontal shift augmentation yields the highest gain, raising the experimental-spectra F1-score from 0.913 to 0.934. This substantially reduces the performance gap induced by the scarcity of real-world experimental data. Whereas combining transformations yields no further gain and can slightly reduce performance, which may suggest that excessive transformation begins to distort chemically meaningful spectral structure.

To validate practical utility, we rigorously evaluated MSpecTmol's resilience to data imperfections common in experimental settings. As illustrated in Fig. 3(e), the model maintains exceptional stability against spectral loss, with F1-scores remaining above 0.86 even when individual modalities are entirely absent. Remarkably, MSpecTmol operating without mass spectrometry still outperforms the full-input 1D-CNN baseline, highlighting the efficacy of our PA-IB architecture in capturing redundant semantic information. This robustness extends to signal interference; as shown in Fig. 3(f), the model exhibits graceful degradation under increasing Gaussian noise, avoiding catastrophic failure even at the most severe perturbation tested ($\sigma = 0.10$, i.e. additive Gaussian noise whose standard deviation reaches 10% of the maximum peak intensity

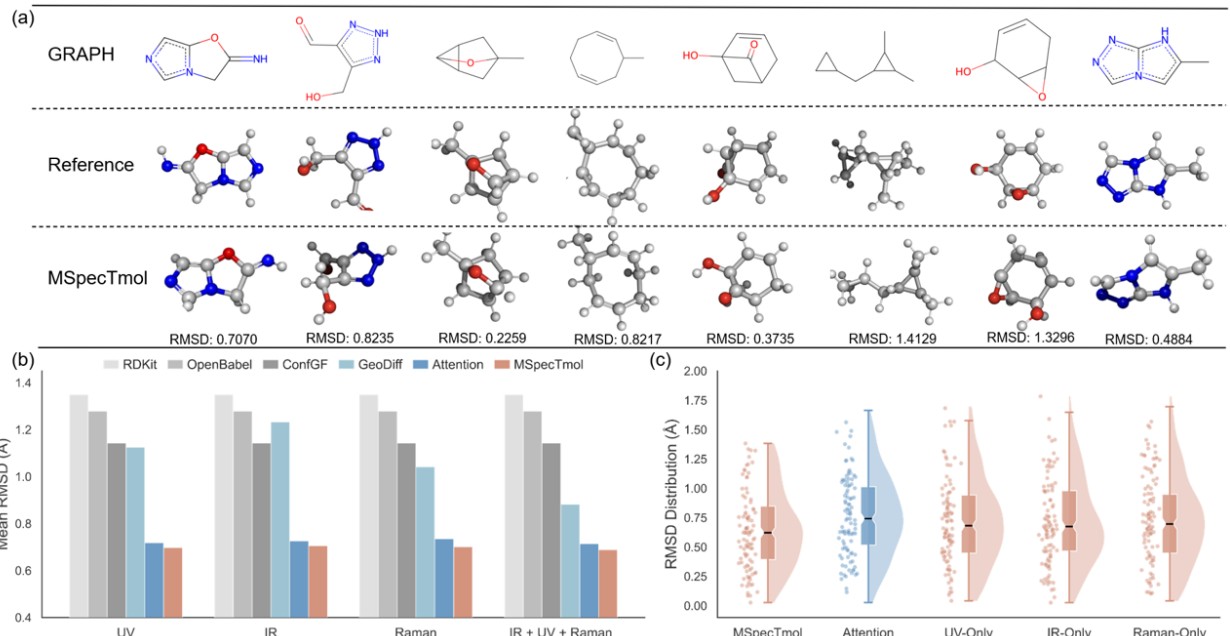

Figure 4: (a) Visualization of generated conformations compared to ground truth for representative molecules. (b) Mean RMSD comparison against spectrum-free (RDKit, OpenBabel, ConfGF) and spectral-conditioned (GeoDiff, Attention) baselines across varying spectral input settings. (c) RMSD distributions comparing MSpecTmol with baseline models.

## 5.3 Results on Spectral-Guided Conformation Generation

To specifically evaluate the stereochemical and spatial information captured by the spectral representations, and to systematically assess the impact of spectral guidance on molecular conformation generation, we compare MSpecTmol with both **spectrum-free graph-based methods** (RDKit, OpenBabel, ConfGF) and **spectral-conditioned baselines**, including an attention-based model and a modified GeoDiff (enhanced to incorporate spectral inputs).

As shown in Figure 4(b), MSpecTmol attains the lowest mean RMSD under *every* spectral configuration, including each single-modality setting. While spectrum-free approaches exhibit limited accuracy and spectral-conditioned baselines show moderate improvements, MSpecTmol remains best across all configurations. As MSpecTmol already operates near the achievable RMSD floor in the single-modality regime, the marginal headroom for further reduction as modalities are added is small; its consistent advantage is therefore best read as dominating at every configuration rather than as a widening gap. That the model genuinely exploits complementary cross-modal information is instead shown by the RMSD distribution in Figure 4(c), where multi-modal fusion yields a substantially lower median and a markedly tighter spread than any single-modality variant—indicating higher consistency, not merely better average accuracy.

The qualitative comparisons in Figure 4(a) further corroborate these quantitative findings. For representative molecules, MSpecTmol produces conformations that closely align with the reference structures. These results illustrate MSpecTmol's ability to translate complex spectral patterns into high-fidelity three-dimensional molecular conformations.

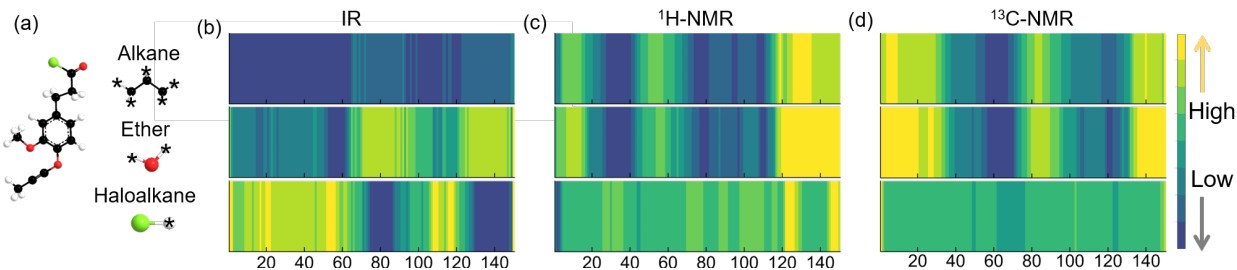

Figure 5: Illustration of spectral information importance. (a) Molecular structure of `CC#COc1ccc(CCC(=O)Cl)cc1OC`, which contains Alkane, Ether, and Haloalkane functional groups. (b) Importance map of IR spectra, (c) $^{13}$C-NMR spectra, (d) and $^{1}$H-NMR spectra. The x-axis of all three spectral plots is normalized to the range [0, 150]. Each map shows the attention scores per position, averaged over the dataset and over the 8 attention heads of the module.

Table 2: Performance comparison (F1-score) between our asymmetric PA-IB and the symmetric Uniform IB baseline across different modality configurations.[2]

| Fusion Strategy | IR | $^{13}$C-NMR | $^{1}$H-NMR | Multi-Modal (All) |
|---|---|---|---|---|
| Symmetric (Uniform IB) | 0.906 | 0.904 | 0.905 | 0.934 |
| **Asymmetric (Ours)** | **0.923** | **0.920** | **0.927** | **0.959** |

In contrast, models conditioned on a single spectral modality (UV-Only, IR-Only, or Raman-Only) exhibit higher median RMSDs and broader distributions, reflecting increased variance in generation quality. A similar degradation is observed in the attention-based ablation model, whose widely dispersed RMSD distribution highlights the limitations of naive fusion strategies. These observations confirm that effective multi-modal fusion is critical for achieving both accurate and stable conformation generation, and they underscore the role of MSpecTmol's fusion architecture in fully exploiting complementary spectral information.

### 5.4 Interpretability and component analysis

To investigate the intrinsic relationships between spectral segments and molecular substructures, as shown in Figure 5, we analyzed the model's attention mechanisms, whose weights have been shown to be chemically interpretable in molecular modeling (Tang et al., 2020; Fine et al., 2020). The visualization reveals that distinct functional groups focus on specific spectral modalities, suggesting that each spectrum encodes information with a unique emphasis. These varying roles underscore the necessity of our PA-IB approach to selectively extract supplementary features from auxiliary modalities while filtering redundancy, with additional visualizations provided in Appendix L.

We further validated the structural advantages of our PA-IB framework by comparing it against a symmetric Uniform IB baseline, which applies identical constraints across all modalities. As presented in Table 2, MSpecTmol consistently outperforms the symmetric approach, achieving a 2.5% absolute gain (0.959 vs. 0.934) in the full multi-modal setting. This confirms that the asymmetric design effectively addresses cross-modal redundancy that symmetric strategies fail to resolve (see Appendix K for detailed fusion strategy comparisons).

We examined the model's sensitivity to the hyperparameters $\alpha$ and $\beta$, which regulate the trade-off between prediction accuracy and information compression in Equation 2. As shown in Figure 6a, setting $\alpha = \beta = 1 \times 10^{-6}$ yields the optimal performance. Larger values impose excessive compression, discarding essential spectral features, while smaller values fail to suppress redundancy, thereby compromising generalization.

---

[2]In the single-modality columns, the auxiliary terms are inactive and the objective reduces to $\mathcal{L}_{\text{pred}} + \alpha\mathcal{L}_{MI^2}$.

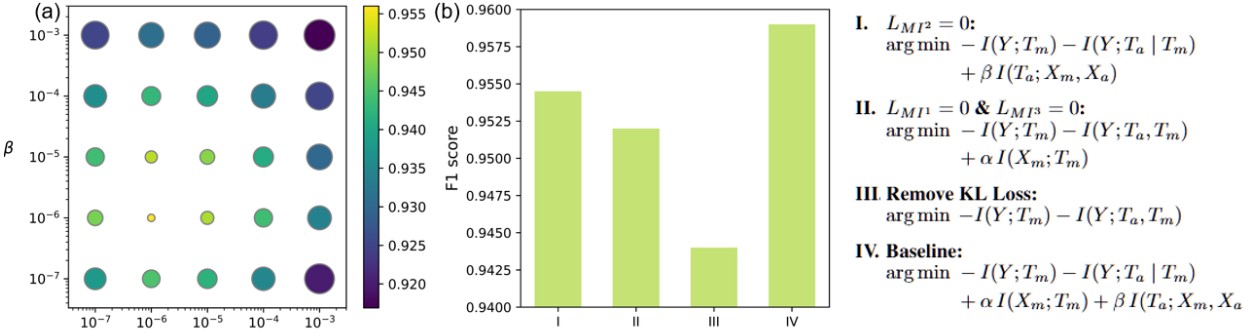

Figure 6: (a) Hyperparameter Experiments on functional group classification task. The circle size is proportional to the magnitude of the error. (b) Ablation study: by selectively removing different KL divergence terms, we adjust the optimization objectives of the model. The left panel shows the F1 scores of the prediction results, while the right panel illustrates the minimized objectives of the ablated models.

Additionally, to quantify the contributions of individual constraints, we conducted ablation studies on the loss terms, as shown in Figure 6b. The full model achieved an F1-score of 0.9589. Removing the KL divergence for auxiliary spectra ($\mathcal{L}_{MI^1}$ and $\mathcal{L}_{MI^3}$) resulted in a substantial decrease to 0.9521, whereas removing the primary spectrum constraint ($\mathcal{L}_{MI^2}$) yielded a smaller drop to 0.9543. This indicates that uncompressed auxiliary information introduces noise that undermines prediction quality, highlighting the importance of regulating information at both levels.

# 6 Conclusion and Future outlook

In this work, we introduce **MSpecTmol**, a multi-modal spectrum information fusion framework based on the information bottleneck principle, designed for automated molecular structure elucidation. Our PA-IB framework adopts a primary-auxiliary synergistic modeling approach, which distills core information from a primary modality while leveraging auxiliary spectra to supplement and refine the final representation. Rigorous experimental evaluations validate MSpecTmol's end-to-end effectiveness, achieving a SOTA F1-score of 0.959 in molecular identification and a low average RMSD of 0.682Å in 3D conformation generation. Meanwhile, our model provides chemically interpretable spectroscopic fragment importance, bridging the gap between ML predictions and domain knowledge.

Looking forward, this framework not only assists chemists in unraveling complex molecular systems but also accelerates the analysis of novel compounds. MSpecTmol holds potential to benefit diverse scientific domains—such as drug discovery, materials science, and chemical forensics—where accurate and reliable molecular identification is critical. MSpecTmol paves the way toward democratized, efficient, and interpretable molecular analysis for broad scientific and industrial applications.

# 7 Reproducibility

We provide the complete implementation in the repository along with guidance on how to reproduce our results. Our code is available at `https://github.com/QXH365/MspecTmol`.

# 8 Ethics Statement

Our study does not involve human participants, personal data, or sensitive information. The datasets and resources used are either publicly available or released under appropriate licenses. We confirm that our research does not raise any ethical concerns related to privacy, safety, fairness, or potential misuse. The contributions of this work are intended solely for advancing scientific research and are not designed or evaluated for harmful applications.

## 9    Acknowledgement

This paper is partially supported by the National Natural Science Foundation of China (No.62502491; No.12227901). The AI-driven experiments, simulations and model training were performed on the robotic AI-Scientist platform of Chinese Academy of Sciences., Anhui Science Foundation for Distinguished Young Scholars (No.1908085J24).

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

## A Training Settings

For the functional group classification task, the model was trained for 100 epochs with a batch size of 128, using the Adam optimizer with an initial learning rate of $4 \times 10^{-4}$ and a cosine annealing scheduler. The loss coefficients $\alpha$ and $\beta$ were set to $1 \times 10^{-6}$, while the weight for the auxiliary prediction loss was 0.7. The temperature for the information bottleneck's stochastic gating was maintained at 1.0. For the spectrum-conditioned molecular conformation generation task, the model was trained for 10,000 iterations with a batch size of 64. We employed the Adam optimizer with a learning rate of $1 \times 10^{-3}$, which was adjusted by a plateau scheduler based on validation loss. For this task, the loss coefficients $\alpha$ and $\beta$ were both set to $1 \times 10^{-6}$. Both models were trained on two NVIDIA A100 GPUs (80 GB each). The classification model required approximately 6 hours of training, while the conformation generation model took around 30 hours.

## B Broader Impacts and Limitation Discussion

### B.1 Broader Impacts

This work promotes automated and interpretable molecular structure elucidation via multi-modal spectroscopic learning. MSpecTmol may assist domains such as drug discovery, materials science, and chemical forensics by providing chemically intuitive insights and reducing reliance on manual spectral interpretation. Its interpretable design supports broader and more accessible molecular analysis. The proposed framework can reduce reliance on extensive manual spectral interpretation, democratizing molecular analysis for broader scientific and industrial use.

### B.2 Limitations

While MSpecTmol demonstrates strong performance, several limitations remain. First, the model's effectiveness depends on the availability of complete, multi-modal spectra, which are often scarce in practice and may hinder its deployment on incomplete datasets. Additionally, its training on a fixed vocabulary of functional groups restricts its ability to identify rare or novel substructures, particularly when analyzing new chemical entities. Future work will focus on addressing these challenges to improve the model's robustness and expand its chemical scope.

## C Proof

### C.1 Proof of Proposition

We first prove the bound on $-I(Y; T_m)$ used in Proposition 3.1. $p_\theta(\mathbf{Y}|T_m)$ is variational approximation of $p(\mathbf{Y}|T_m)$. We model $p_\theta(\mathbf{Y}|T_m)$ as a predictor parametrized by $\theta$, which outputs the model prediction $\mathbf{Y}$ based on the core primary spectra $T_m$.

$$
\begin{aligned}
I(\mathbf{Y}; T_m) &= \mathbb{E}_{\mathbf{Y}, T_m}[\log \frac{p(\mathbf{Y}|T_m)}{p(\mathbf{Y})}] \\
&= \mathbb{E}_{\mathbf{Y}, T_m}[\log \frac{p_\theta(\mathbf{Y}|T_m)}{p(\mathbf{Y})}] \\
&\quad + \mathbb{E}_{T_m}[KL(p(\mathbf{Y}|T_m)||p_\theta(\mathbf{Y}|T_m))]
\end{aligned}
\tag{15}
$$

According to the non-negativity of the KL divergence, we have:

$$
\begin{aligned}
I(\mathbf{Y}; T_m) &\geq \mathbb{E}_{\mathbf{Y}, T_m}[\log \frac{p_\theta(\mathbf{Y}|T_m)}{p(\mathbf{Y})}] \\
&= \mathbb{E}_{\mathbf{Y}, T_m}[\log p_\theta(\mathbf{Y}|T_m)] + H(\mathbf{Y})
\end{aligned}
\tag{16}
$$

Thus, we can minimize the upper bound of $-I(\mathbf{Y}; T_m)$ by minimizing the model prediction loss $\mathcal{L}_{\mathrm{pred}}(\mathbf{Y}, T_m)$, which can be modeled as the cross entropy loss for classification and the mean square loss for regression.

### C.1.1 Minimizing $-I\big(Y;T_a \mid T_m\big)$

For the second term of Equation **??**, i.e., $-I\big(Y;T_a \mid T_m\big)$, we decompose the term into the sum of two terms based on the chain rule of mutual information as follows:

$$I(Y;T_a|T_m) = I(Y;T_a,T_m) - I(T_a;T_m). \tag{17}$$

For the upper bound of $-I(Y;T_a,T_m)$, Given the core primary spectra $T_m$ and core auxiliary spectra $T_a$, and its label information $\mathbf{Y}$, we have

$$
\begin{aligned}
-I(\mathbf{Y};T_a,T_m) &\leq \mathbb{E}_{(T_a,T_m,\mathbf{Y})}[-\log p_\theta(\mathbf{Y}|T_a,T_m)] \\
&= \mathbb{E}_{(\mathbf{Y},T_a,T_m)} \log\left[P_\theta\left(\mathbf{Y} \mid T_a, T_m\right)\right] + H(\mathbf{Y}) := \mathcal{L}_{sup},
\end{aligned}
\tag{18}
$$

where $p_\theta(\mathbf{Y}|T_a,T_m)$ is variational approximation of $p(\mathbf{Y}|T_a,T_m)$. We model $p_\theta(\mathbf{Y}|T_a,T_m)$ as a predictor parametrized by $\theta$, which outputs the model prediction $\mathbf{Y}$ based on the core spectra $T_a$ and $T_m$. Thus, we can minimize the upper bound of $-I(\mathbf{Y};T_a,T_m)$ by minimizing the supplementary prediction loss $\mathcal{L}_{\text{sup}}$,

For the upper bound of $I(T_a;T_m)$, drawing inspiration from the experiences derived in Variational Autoencoders (VAE) (Kingma, 2013), we attempt to replace $p(t_a)$ with $q(t_a)$ and consolidate the additional components to form a Kullback-Leibler (KL) divergence:

$$
\begin{aligned}
I(T_a;T_m) &= \mathbb{E}_{(t_m,t_a)\sim p(t_m,t_a)}\left[\log \frac{p(t_a \mid t_m)}{p(t_a)}\right] \\
&= \mathbb{E}_{(t_m,t_a)\sim p(t_m,t_a)}\left[\log \frac{p(t_a \mid t_m)}{q(t_a)} \cdot \frac{q(t_a)}{p(t_a)}\right] \\
&= \mathbb{E}_{(t_m,t_a)\sim p(t_m,t_a)}\left[\log \frac{p(t_a \mid t_m)}{q(t_a)}\right] \\
&\quad + \mathbb{E}_{(t_m,t_a)\sim p(t_m,t_a)}\left[\log \frac{q(t_a)}{p(t_a)}\right]
\end{aligned}
\tag{19}
$$

For the first term, both $p(t_a \mid t_m)$ and $q(t_a)$ have analytical forms, allowing the function within the brackets to be computed analytically. By utilizing the relationship $p(t_m,t_a) = p(t_m)p(t_a \mid t_m)$, we can rewrite the first term in a more elegant manner:

$$
\begin{aligned}
&\mathbb{E}_{(t_m,t_a)\sim p(t_m,t_a)}\left[\log \frac{p(t_a \mid t_m)}{q(t_a)}\right] \\
&= \iint p(t_m)p(t_a \mid t_m)\log \frac{p(t_a \mid t_m)}{q(t_a)}\, \mathrm{d}t_a\, \mathrm{d}t_m \\
&= \int p(t_m)\left(\int p(t_a \mid t_m)\log \frac{p(t_a \mid t_m)}{q(t_a)}\, \mathrm{d}t_a\right)\mathrm{d}t_m \\
&= \mathbb{E}_{t_m\sim p(t_m)}\left[\mathrm{KL}\left(p(t_a \mid t_m)\|q(t_a)\right)\right] \\
&:= L_{MI^1} \approx \frac{1}{N}\sum_{i=1}^{N}\mathrm{KL}\left[p\left(t_a \mid t_{mi}\right)\|q(t_a)\right], \quad t_{mi} \sim p(t_m)
\end{aligned}
\tag{20}
$$

The term $L_{MI1} := \mathbb{E}_{t_m\sim p(t_m)}[\mathrm{KL}(p(t_a \mid t_m)\|q(t_a))]$ is often referred to as the *rate* in rate-distortion theory. This rate component can be optimized using mini-batch gradient descent. Specifically, by sampling a batch of training samples $t_{m1},\ldots,t_{mN}$ from the training set, we can minimize the KL divergence $\mathrm{KL}\left[p\left(t_a \mid t_{mi}\right)\|q(t_a)\right]$ for each $t_{mi}$.

Since both distributions $p(t_a \mid t_m)$ and $q(t_a)$ are Gaussian, the KL divergence between them has an analytical solution:

$$
\begin{aligned}
&\mathrm{KL}\left[p(t_a \mid t_m)\|q(t_a)\right] \\
&= \mathrm{KL}\left[\mathcal{N}\left(\boldsymbol{\mu}(t_m), \boldsymbol{\sigma}^2(t_m)\boldsymbol{I}\right) \| \mathcal{N}(\boldsymbol{0}, \boldsymbol{I})\right] \\
&= \sum_{j=1}^{J} \mathrm{KL}\left[\mathcal{N}\left(\mu_j, \sigma_j^2\right) \| \mathcal{N}(0, 1)\right] \\
&= \sum_{j=1}^{J} \frac{1}{2}\left(-\log \sigma_j^2 - 1 + \mu_j^2 + \sigma_j^2\right)
\end{aligned}
\tag{21}
$$

Here, $\mu(t_m), \sigma^2(t_m)$ are mean and variance of $\mathbf{H}^m$, respectively.

### C.1.2 Minimizing $- I\left(X_m; T_m\right)$

For the upper bound of $- I\left(X_m; T_m\right)$, Similarly, we attempt to replace $p(t_m)$ with $q(t_m)$ and consolidate the additional components to form a Kullback-Leibler (KL) divergence:

$$
\begin{aligned}
I(X_m; T_m) = \ &\mathbb{E}_{(t_m, x_m) \sim p(t_m, x_m)}\left[\log \frac{p(t_m \mid x_m)}{q_{(t_m)}}\right] \\
&+ \mathbb{E}_{(t_m, x_m) \sim p(t_m, x_m)}\left[\log \frac{q(t_m)}{p(t_m)}\right]
\end{aligned}
\tag{22}
$$

By utilizing the relationship $p(t_m, x_m) = p(x_m)p(t_m \mid x_m)$, we can rewrite the first term in a more elegant manner:

$$
\begin{aligned}
&\mathbb{E}_{(t_m, x_m) \sim p(t_m, x_m)}\left[\log \frac{p(t_m \mid x_m)}{q(t_m)}\right] \\
&= \mathbb{E}_{t_m \sim p(t_m)}\left[\mathrm{KL}\left(p(t_m \mid x_m)\|q(t_m)\right)\right] := L_{MI^2}
\end{aligned}
\tag{23}
$$

As with $L_{MI^1}$, the term $L_{MI^2}$ is the *rate* term in rate-distortion theory and is optimized via mini-batch gradient descent. Since both $p(t_m \mid x_m)$ and $q(t_m)$ are Gaussian, it admits the same analytical KL solution:

$$
\mathrm{KL}\left[p(t_m \mid x_m)\|q(t_m)\right] = \sum_{j=1}^{J} \frac{1}{2}\left(-\log \sigma_j^2 - 1 + \mu_j^2 + \sigma_j^2\right)
\tag{24}
$$

### C.1.3 Minimizing $- I\left(T_a; X_m, X_a\right)$

For the upper bound of $- I\left(T_a; X_m, X_a\right)$, Similarly, we attempt to replace $p(t_a)$ with $q(t_a)$ and consolidate the additional components to form a Kullback-Leibler (KL) divergence:

$$
\begin{aligned}
I(T_a; X_m, X_a) = \ &\mathbb{E}_{(t_a, x_a, x_m) \sim p(t_a, x_a, x_m)}\left[\log \frac{p(t_a \mid x_m, x_a)}{q_{(t_a)}}\right] \\
&+ \mathbb{E}_{(t_a, x_a, x_m) \sim p(t_a, x_a, x_m)}\left[\log \frac{q(t_a)}{p(t_a)}\right]
\end{aligned}
\tag{25}
$$

By utilizing the relationship $p(t_a, x_a, x_m) = p(t_a, x_a)p(t_a \mid x_m, x_a)$, we can rewrite the first term in a more elegant way:

$$
\begin{aligned}
&\mathbb{E}_{(t_a, x_a, x_m) \sim p(t_a, x_a, x_m)}\left[\log \frac{p(t_a \mid x_m, x_a)}{q(t_a)}\right] \\
&= \mathbb{E}_{t_a, x_a \sim p(x_m, x_a)}\left[\mathrm{KL}\left(p(t_a \mid x_m, x_a)\|q(t_a)\right)\right] := L_{MI^3}
\end{aligned}
\tag{26}
$$

Analogously, $L_{MI^3}$ is the *rate* term and is optimized in the same manner; since both $p(t_a \mid x_m, x_a)$ and $q(t_a)$ are Gaussian, the KL again has the closed form:

$$
\mathrm{KL}\left[p(t_a \mid x_m, x_a)\|q(t_a)\right] = \sum_{j=1}^{J} \frac{1}{2}\left(-\log \sigma_j^2 - 1 + \mu_j^2 + \sigma_j^2\right)
\tag{27}
$$

# D  Definition of Functional Groups

Functional groups play a crucial role in determining the chemical reactivity and properties of molecules. To systematically analyze molecular structures, we employ a set of predefined patterns to identify key functional groups within a given molecular dataset.

Table 3 lists the functional groups considered in this study, along with their corresponding SMARTS representations. These functional groups were selected based on their relevance to organic and medicinal chemistry, including common moieties such as hydroxyl (-OH), carbonyl (C=O), and amine ($-NH_2$) groups. The identification process involves scanning molecular structures using subgraph matching algorithms, ensuring accurate detection of these structural motifs.

Table 3: Predefined Functional Groups and Their SMARTS Patterns

| Functional Group | SMARTS Pattern |
|---|---|
| Acid anhydride | `[CX3](=[OX1])[OX2][CX3](=[OX1])` |
| Acyl halide | `[CX3](=[OX1])[F,Cl,Br,I]` |
| Alcohol | `[#6][OX2H]` |
| Aldehyde | `[CX3H1](=O)[#6,H]` |
| Alkane | `[CX4;H3,H2]` |
| Alkene | `[CX3]=[CX3]` |
| Alkyne | `[CX2]#[CX2]` |
| Amide | `[NX3][CX3](=[OX1])[#6]` |
| Amine | `[NX3;H2,H1,H0;!$(NC=O)]` |
| Arene | `[cX3]1[cX3][cX3][cX3][cX3][cX3]1` |
| Azo compound | `[#6][NX2]=[NX2][#6]` |
| Carbamate | `[NX3][CX3](=[OX1])[OX2H0]` |
| Carboxylic acid | `[CX3](=O)[OX2H]` |
| Enamine | `[NX3][CX3]=[CX3]` |
| Enol | `[OX2H][#6X3]=[#6]` |
| Ester | `[#6][CX3](=O)[OX2H0][#6]` |
| Ether | `[OD2]([#6])[#6]` |
| Haloalkane | `[#6][F,Cl,Br,I]` |
| Hydrazine | `[NX3][NX3]` |
| Hydrazone | `[NX3][NX2]=[#6]` |
| Imide | `[CX3](=[OX1])[NX3][CX3](=[OX1])` |
| Imine | `[$([CX3]([#6])[#6]),$([CX3H][#6])]=[$([NX2][#6]),$([NX2H])]` |
| Isocyanate | `[NX2]=[C]=[O]` |
| Isothiocyanate | `[NX2]=[C]=[S]` |
| Ketone | `[#6][CX3](=O)[#6]` |
| Nitrile | `[NX1]#[CX2]` |
| Phenol | `[OX2H][cX3]:[c]` |
| Phosphine | `[PX3]` |
| Sulfide | `[#16X2H0]` |
| Sulfonamide | `[#16X4]([NX3])(=[OX1])(=[OX1])[#6]` |
| Sulfonate | `[#16X4](=[OX1])(=[OX1])([#6])[OX2H0]` |
| Sulfone | `[#16X4](=[OX1])(=[OX1])([#6])[#6]` |
| Sulfonic acid | `[#16X4](=[OX1])(=[OX1])([#6])[OX2H]` |
| Sulfoxide | `[#16X3]=[OX1]` |
| Thial | `[CX3H1](=S)[#6,H]` |
| Thioamide | `[NX3][CX3]=[SX1]` |
| Thiol | `[#16X2H]` |

The functional group identification is performed using cheminformatics libraries such as RDKit, which allows for efficient substructure searches within molecular datasets. This approach enables us to extract chemically meaningful information and facilitate downstream tasks such as molecular property prediction, reactivity analysis, and structure-based clustering.

# E  Complexity Analysis and Choice of Primary Spectra

We conducted a comprehensive analysis to determine the optimal multi-modal configuration for MSpecTmol, balancing predictive performance with computational efficiency. The time and space complexity of our model and several baselines are presented in Table 4. This analysis reveals a clear trade-off between the number of input modalities and the required resources. As shown in table 5, while expanding from three to five spectral inputs nearly doubled the resource consumption, it yielded only marginal performance improvements. This finding led us to select a three-modality fusion as the most balanced and efficient configuration.

A critical aspect of our framework is the selection of the primary spectrum, as MSpecTmol is designed to prioritize its features while using auxiliary spectra for supplementary information. Our recommended procedure is to first identify the single best-performing modality in standalone experiments and assign it the primary role, thereby ensuring that the most informative stream is preserved.

To implement this strategy for the functional group classification task, we first assessed the predictive power of each individual spectrum (Table 1 and Figure 9(a)). The results revealed that IR spectroscopy delivered relatively high and stable accuracy, making it the ideal candidate for the primary spectrum. Conversely, MS/MS spectra exhibited the lowest performance. In our multi-modal evaluations, we observed that fusing multiple spectra consistently improved performance. Notably, the combination of IR, $^1$H-NMR, and $^{13}$C-NMR not only outperformed other fusion strategies—achieving the best results for 35 out of 37 functional groups—but was also more effective than using all available spectra, all while maintaining lower computational complexity (Figure 9(b)). Consequently, we established the optimal configuration for this task as using IR spectroscopy as the primary input, with the $^1$H-NMR and $^{13}$C-NMR modalities serving as powerful auxiliary inputs.

Table 4: Comparison of resource usage and performance.

| Model | Mem.(GB) | Time(h) | F1-score |
|---|---|---|---|
| 1D-CNN | 5.7 | 2 | 0.900 |
| Transformer | 1.7 | 9 | 0.911 |
| MSpecTmol | 6 | 3 | 0.959 |

Table 5: Comparison across different numbers of modalities.

| #Mod. | Mem.(GB) | Time(h) | F1-score |
|---|---|---|---|
| 1 | 3.4 | 2 | 0.923 |
| 3 | 6.6 | 2.5 | 0.959 |
| 5 | 11.3 | 5 | 0.963 |

Table 6: Inference time for processing 10,000 samples.

| Model | Inference Time (s) |
|---|---|
| 1D-CNN | 10.4 |
| Transformer | 45.1 |
| Wu et al. | 55.4 |
| Alberts et al. | 60.1 |
| **MSpecTmol** | **14.0** |

Table 7: Training time vs. Molecular Size (Heavy Atom Count).

| Heavy Atom Count | Training Time (s) |
|---|---|
| 5 - 15 | 564 |
| 16 - 25 | 556 |
| 25 - 35 | 558 |

# F  Inference Time and Scalability

**Inference Time**  To assess the model's suitability for high-throughput screening, we measured the total inference time for processing 10,000 samples on a single NVIDIA A100 GPU. Table 6 shows that MSpecTmol completes the task in just 14.0 seconds. This speed is comparable to the simple 1D-CNN (10.4 s) and drastically faster than the Transformer (45.1s), confirming its efficiency for real-time applications.

**Scalability with Molecular Size**  To verify whether the model's computational cost is sensitive to molecular complexity, we measured the training time on subsets of data sorted by Heavy Atom Coun. As presented

in Table 7, the training time remains remarkably consistent ($\approx 560s$) across different molecular sizes. This is because our model takes fixed-dimension interpolated spectra as input and outputs functional group probabilities; consequently, the physical size or complexity of the molecule does not alter the input tensor dimensions or the model architecture.

## G Comparison with LLM-based Baselines

We evaluated MSpecTmol by strictly adopting the experimental settings of SpectraLLM, focusing on the classification of the same 17 specific functional groups. As shown in Table G, MSpecTmol consistently outperforms SpectraLLM across all datasets and modality configurations, indicating a more effective inductive bias for structured spectroscopic discriminative tasks.

Table 8: Performance comparison between SpectraLLM and MSpecTmol.

| Dataset | Modality | SpectraLLM | MSpecTmol (Ours) |
|---------|----------|------------|------------------|
| QM9S | IR | 0.659 | **0.970** |
| QM9S | Raman | 0.731 | **0.966** |
| QM9S | UV-Vis | 0.371 | **0.683** |
| QM9S | IR + Raman + UV-Vis | 0.793 | **0.982** |
| Alberts | IR | 0.602 | **0.935** |
| Alberts | $^{13}$C-NMR | 0.424 | **0.927** |
| Alberts | $^{1}$H-NMR | 0.332 | **0.925** |
| Alberts | IR + NMR | 0.776 | **0.970** |
| Alberts | IR + MS | 0.634 | **0.953** |

## H Sensitivity Analysis of Gumbel-Softmax Temperature

In the Core Spectrum Extraction module, we employ Gumbel-Softmax to enable differentiable sampling of the discrete importance masks. The temperature parameter $t$ plays a pivotal role in controlling the sharpness of this distribution. To evaluate its impact on model performance, we conducted a sensitivity analysis across various temperatures. As shown in Table 9, MSpecTmol achieves the optimal F1-score at $t = 1.0$. When the temperature is set to a lower value ($t = 0.5$), the performance experiences a slight decline. Conversely, increasing the temperature to higher values ($t = 1.5, 2.0$) leads to a more noticeable degradation in prediction quality.

Table 9: Sensitivity analysis of the Gumbel-Softmax temperature parameter $t$.

| Temperature ($t$) | F1-score |
|-------------------|----------|
| 0.5 | 0.952 |
| **1.0** | **0.959** |
| 1.5 | 0.954 |
| 2.0 | 0.945 |

Intuitively, the temperature controls the sparsity and sharpness of the gating over spectral frequency bands. In our framework, each gate determines whether a local region of the spectrum is preserved or replaced by noise. A very low temperature makes these decisions almost binary. While this promotes sparsity, it risks discarding weak but chemically informative peaks (e.g., small shoulders or minor bands) that are critical for distinguishing fine-grained functional groups and isomers. On the other hand, a high temperature yields overly soft gates, causing most bands to be partially retained. This weakens the model's ability to suppress redundancy and blurs the importance patterns across modalities. The superior performance observed at $t = 1.0$ confirms that this setting achieves an optimal balance, allowing the PA-IB framework to learn selective yet stable masks that retain structurally informative spectral regions while effectively filtering out redundant or noisy segments.

# I  Generalization Analysis on Complex Molecular Structures

To evaluate the model's generalization capability on samples with more complex molecular structures and denser, overlapping spectral peaks, we performed a supplementary stress test on the dataset from Alberts et al. Specifically, instead of a standard random split, we sorted the entire dataset by heavy atom count. We utilized the bottom 90% for training and reserved the top 10% strictly for testing. This setup introduces a significant distribution shift, requiring the model to infer the structure of complex molecules that are physically larger than any sample seen during training.

The results are presented in Table 10. While the performance on unseen larger molecules naturally dips compared to the standard random split due to the increased structural complexity, MSpecTmol maintains a high F1-score of 0.925. Notably, our model consistently outperforms baselines in this challenging setting. This demonstrates that our PA-IB framework effectively learns intrinsic spectroscopic-structural correlations rather than simply memorizing dataset-specific patterns, confirming its capability to generalize to more complex chemical spaces.

Table 10: Performance comparison (F1-score) on the stress test of the top 10% largest molecules versus the original random split.

| Modality | Model | f1-score (Top 10% Large) | f1-score (Original Split) |
|---|---|---|---|
| IR | 1D-CNN | 0.866 | 0.895 |
| | Transformer | 0.852 | 0.881 |
| | Wu et al. | 0.864 | 0.886 |
| | Alberts et al. | 0.874 | 0.891 |
| | **MSpecTmol** | **0.900** | **0.920** |
| $^{13}$C-NMR | 1D-CNN | 0.623 | 0.674 |
| | Transformer | 0.845 | 0.913 |
| | Wu et al. | 0.873 | 0.914 |
| | Alberts et al. | 0.896 | 0.919 |
| | **MSpecTmol** | **0.904** | **0.923** |
| IR + $^{13}$C-NMR + $^{1}$H-NMR | 1D-CNN | 0.873 | 0.900 |
| | Transformer | 0.902 | 0.936 |
| | Wu et al. | 0.912 | 0.944 |
| | Alberts et al. | 0.916 | 0.947 |
| | **MSpecTmol** | **0.925** | **0.959** |

# J  Analysis of Prior Distribution Choice for Latent Variables

In our Primary-Auxiliary Information Bottleneck (PA-IB) framework, the choice of the prior distributions for the latent bottleneck variables $T_m$ and $T_a$, denoted as $q(t_m)$ and $q(t_a)$ respectively, is a critical step that influences model performance. Specifically, $q(t_m)$ regularizes the core information extracted from the primary spectrum, while $q(t_a)$ regularizes the supplementary information from the auxiliary spectra. To ensure the scientific rigor and optimality of our model's configuration, we systematically investigated the impact of different prior distributions on performance.

We designed a series of rigorous comparative experiments to evaluate three distinct prior distributions on the functional group classification task, applying them to both $q(t_m)$ and $q(t_a)$:

1. **Gaussian Distribution**: The standard $\mathcal{N}(0, I)$ distribution.
2. **Laplace Distribution**: The standard $Laplace(0, 1)$ distribution, which is known to effectively promote sparsity in the latent space.
3. **Gamma Distribution**: The standard $\Gamma(k = 1, \theta = 1)$ distribution, which constrains the latent variables to be non-negative.

Throughout these experiments, all other model hyperparameters (such as learning rate, batch size, and the trade-off coefficients $\alpha$ and $\beta$) were held strictly constant to ensure a fair comparison.

The model trained with the **Gaussian prior** achieved the highest F1-score of 0.959, compared to 0.951 for the Laplace prior and 0.946 for the Gamma prior. This superior performance suggests that assuming the compressed latent features of both primary and auxiliary spectra follow a Gaussian distribution provides an efficient and flexible representation space, allowing the model to optimally capture the complex relationships within the spectral data. Therefore, we selected the Gaussian distribution for our final model configuration, as its effectiveness is validated by these results.

## K    Comparison with Alternative Multi-Modal Fusion Strategies

To validate the effectiveness of PA-IB as a multi-modal information fusion strategy, we conducted a comparative study against three standard fusion paradigms:

- **Early Fusion:** The features from all modalities are directly concatenated at the input level before being fed into the model, allowing the model to learn a joint representation from the raw data.

- **Mid-level Fusion:** Each modality is first processed through independent CNN encoders to extract latent features. These features are then concatenated and passed to a shared MLP for classification.

- **Late Fusion:** Each modality independently predicts the presence of functional groups through separate MLPs, and the final prediction is obtained by averaging the probability outputs from all modalities.

The results, presented in Table 11, show that MSpecTmol consistently outperforms all three baselines (in terms of F1-score). Early and Mid-level fusion strategies generally perform better than Late fusion, likely because they enable some degree of joint representation learning. However, they still fall short of MSpecTmol, as they fail to explicitly filter out redundant cross-modal information. Late fusion performs the worst, as it ignores inter-modal interactions entirely by processing each modality in isolation, leading to substantial information loss.

Table 11: Performance comparison (F1-score) of MSpecTmol against Early, Mid-level, and Late fusion strategies across different modality combinations.

| Modality Configuration | MSpecTmol | Early Fusion | Mid-level Fusion | Late Fusion |
|---|---|---|---|---|
| IR + $^{13}$C-NMR + $^{1}$H-NMR | **0.959** | 0.900 | 0.904 | 0.874 |
| IR + MS/MS (Pos) + MS/MS (Neg) | **0.944** | 0.887 | 0.895 | 0.854 |

## L    Interpretability Analysis

To disentangle overlapping importance regions caused by functional group co-occurrence, we adopt a one-vs-all training strategy by training a dedicated model for each functional group. Each model receives the concatenation of all three spectral modalities as input. This design allows us to isolate the contribution of individual spectra to specific functional group predictions and analyze their region-wise importance, as shown in Figure 7.

Different spectral modalities emphasize distinct molecular features. Different types of spectroscopy capture different aspects of molecular structures. Infrared (IR) spectroscopy is particularly important in identifying functional groups such as carbonyl (C=O), hydroxyl (-OH), and amine ($-NH_2$). This is likely because IR spectroscopy primarily reflects the vibrational characteristics of polar functional groups, which exhibit strong absorption in the IR spectrum. In contrast, nuclear magnetic resonance (NMR) spectroscopy is more sensitive to structural motifs such as alkyl ($-CH_3$, $-CH_2-$), aromatic rings, and heterocycles. This is because

NMR provides detailed insights into the electronic environment surrounding specific atomic nuclei, allowing for precise differentiation of these structural features. The complementary nature of these spectral modalities underscores the necessity of multimodal approaches for comprehensive molecular characterization.

## M   Confusion matrix Analysis

To investigate functional group misclassification, we construct a confusion matrix based on co-occurring prediction errors, counting the instances where two groups are simultaneously mispredicted for each test sample. As shown in Figure 8, this reveals that certain groups—notably Ether, Haloalkane, and Sulfide—are frequently confused. To diagnose this, we visualized the model's attention across the fused multi-modal spectra (Figure 7). The analysis demonstrates that these confusable groups exhibit significant overlapping attention, indicating that the model relies on shared features present across different spectral modalities for their identification. This finding highlights the inherent difficulty in distinguishing these groups, even when multiple sources of spectral information are available.

The observed spectral feature overlap is rooted in the intrinsic chemical properties of these functional groups. Similarities in their responses across various spectroscopic methods, such as shared absorption bands or related electronegativity profiles, create highly correlated features that are challenging to disentangle. This inherent ambiguity confirms that no single spectral modality contains sufficient information for perfect discrimination. It therefore becomes critical to employ a framework that can synergistically fuse complementary information from multiple spectra. Our approach is designed to address this very challenge, resolving ambiguities by integrating diverse spectral evidence to achieve more accurate classification.

## N   Impact of Data Augmentation

Real-world experimental spectra are often subject to variations from instrumental noise and calibration drift. This challenge is compounded by the scarcity of large-scale, multi-modal spectral datasets. Given that deep learning models, particularly those with convolutional neural network (CNN) architectures, are inherently data-hungry, data augmentation becomes a crucial technique. By synthetically expanding the training dataset to represent a wider range of experimental conditions, we can significantly enhance the model's generalization, robustness, and overall predictive performance. All experiments were conducted on molecular data obtained from the SDBS database.

We implemented and tested several augmentation strategies with the following specific configurations:

- **Horizontal Shift:** A random horizontal shift of up to 10 pixels is applied to the spectrum's data points.
- **Vertical Noise:** Uniform random noise (up to a level of 0.05) is added to the intensity values, with the noise magnitude being inversely scaled by the signal intensity.
- **Gaussian Smoothing:** A 1D Gaussian filter with a sigma value randomly chosen between 0.75 and 1.25 is applied to the spectrum.
- **Combined Strategies:** A horizontal shift or vertical noise is first applied, followed by the application of Gaussian smoothing.

As shown in Figure 3(d) in the main text, all data augmentation strategies successfully improved the F1-score compared to the model trained on the original data (0.9134). Among these, the Horizontal Shift strategy was the most effective, achieving the highest F1-score of 0.9344. This suggests that teaching the model to be robust against positional variations in spectral peaks is highly beneficial. The addition of vertical noise also provided a substantial performance boost. Interestingly, while Gaussian smoothing alone offered a modest improvement, combining it with other methods (e.g., vertical noise + smoothing) did not yield further gains and resulted in lower performance than the individual, more effective strategies. This indicates that while introducing variability is beneficial, excessive transformation can risk distorting the essential chemical information within the spectra.

## O   Impact of Missing Modalities and Noise Injection

**Impact of Missing Modalities**   To evaluate MSpecTmol's robustness against incomplete data, a common real-world challenge, we masked individual spectral modalities in the test set and assessed the pre-trained model's performance on the SDBS dataset. As presented in Table 12, MSpecTmol's performance degrades gracefully rather than failing: the F1-score drops from 0.9134 with complete data to between 0.8974 and 0.8623 when a modality is absent. This resilience, a direct benefit of our PA-IB architecture, allows MSpecT-mol to still outperform several baselines operating with complete data. Its ability to extract information from auxiliary spectra to supplement the primary modality enables robust performance even when some data are missing.

Table 12: Performance comparison (F1-score) under missing modality conditions.

| Input Configuration | MSpecTmol | 1D-CNN | Trans. | Wu et al. | Alberts et al. |
|---|---|---|---|---|---|
| Full (MS+$^{13}$C+$^1$H) | **0.913** | 0.847 | 0.858 | 0.872 | 0.881 |
| w/o MS | **0.862** | 0.811 | 0.826 | 0.831 | 0.847 |
| w/o $^1$H-NMR | **0.897** | 0.823 | 0.834 | 0.848 | 0.866 |
| w/o $^{13}$C-NMR | **0.878** | 0.819 | 0.818 | 0.832 | 0.851 |

**Impact of Noise**   Noise is an unavoidable component of experimental spectra. To assess this, we introduced varying levels of Gaussian noise to all spectra in the test set. The standard deviation of the noise was scaled proportionally to the maximum intensity of each spectrum. As illustrated in Table 13, the model's F1-score exhibits a steady and predictable decline as the noise level increases, decreasing from 0.9134 on clean data to 0.7469 at the highest noise level of 0.1. Importantly, the performance does not suffer a catastrophic collapse but rather degrades gracefully. At the highest noise level ($\sigma = 0.10$), MSpecTmol (0.747) significantly outperforms 1D-CNN (0.635) and Transformer (0.704). This demonstrates that MSpecTmol can effectively discern core spectral features from background noise.

Table 13: Performance comparison (F1-score) under varying levels of Gaussian noise.

| Noise ($\sigma$) | MSpecTmol | 1D-CNN | Trans. | Wu et al. | Alberts et al. |
|---|---|---|---|---|---|
| 0.00 (Clean) | **0.913** | 0.847 | 0.858 | 0.872 | 0.881 |
| 0.02 | **0.882** | 0.815 | 0.825 | 0.838 | 0.846 |
| 0.05 | **0.825** | 0.745 | 0.761 | 0.767 | 0.772 |
| 0.10 | **0.747** | 0.635 | 0.704 | 0.711 | 0.703 |

## P   Algorithmic Procedure of Conformation Generation

In this work, we propose a dual-encoder diffusion framework that generates molecular conformations by conditioning a geometric diffusion model on spectroscopic information.

### P.1   Problem Formulation

Given a molecular graph $G = (\mathcal{V}, \mathcal{E})$ with atom types $\mathbf{z} \in \mathbb{Z}^{|\mathcal{V}|}$ and spectroscopic measurements $\mathbf{s} = [\mathbf{s}_{uv}, \mathbf{s}_{ir}, \mathbf{s}_{raman}] \in \mathbb{R}^{d_s}$, we aim to generate 3D molecular conformations $\mathbf{x} \in \mathbb{R}^{3|\mathcal{V}|}$ that are consistent with both the molecular connectivity and observed spectra.

### P.2   Spectrum-Conditioned Diffusion Process

We formulate the generation process as a conditional diffusion model operating in the coordinate space. The forward diffusion process adds Gaussian noise to the true conformation $\mathbf{x}_0$:

$$q(\mathbf{x}_t|\mathbf{x}_0) = \mathcal{N}(\mathbf{x}_t; \sqrt{\alpha_t}\mathbf{x}_0, (1 - \alpha_t)\mathbf{I}) \tag{28}$$

where $\alpha_t = \prod_{i=1}^{t}(1 - \beta_i)$ and $\{\beta_i\}$ follows a predefined noise schedule.

The reverse process is parameterized by a neural network $\epsilon_\theta$ that predicts the noise conditioned on the spectrum:

$$\mathbf{x}_{t-1} = \frac{1}{\sqrt{1 - \beta_t}} \left( \mathbf{x}_t - \frac{\beta_t}{\sqrt{1 - \alpha_t}} \epsilon_\theta(\mathbf{x}_t, t, \mathbf{s}, G) \right) + \sigma_t \boldsymbol{\eta} \tag{29}$$

where $\boldsymbol{\eta} \sim \mathcal{N}(0, \mathbf{I})$ and $\sigma_t$ is the posterior variance.

### P.3 Dual-Encoder Architecture

Our model consists of three key components:

**Spectrum Encoder:** We design a MSpecTmol encoder to process multi-modal spectroscopic data. The spectrum data is encoded using the PA-IB-based method described previously. The features are fused through a gated mechanism:

$$\mathbf{h}_s = \mathrm{MLP}(\mathbf{h}_{spec} \oplus \mathbf{t}_{emb}) \tag{30}$$

where $\mathbf{h}_{spec}$ is the spectrum embedding, $\mathbf{t}_{emb}$ is the timestep embedding, and $\oplus$ denotes concatenation.

**Dual Geometric Encoders:** We employ two complementary graph encoders: (1) a SchNet-based global encoder that captures long-range interactions through radius graphs, and (2) a GIN-based local encoder focusing on chemical bond structures. Both encoders incorporate the spectrum condition $\mathbf{h}_s$ into node representations.

**Distance-based Denoising:** In this step, we predict noise in the distance space and transform back to coordinates. The training objective combines global and local distance predictions:

$$\mathcal{L} = \mathbb{E}_{t,\epsilon} \left[ \lambda_g \|\mathbf{d}_g - \hat{\mathbf{d}}_g\|_2^2 + \lambda_l \|\mathbf{d}_l - \hat{\mathbf{d}}_l\|_2^2 \right] \tag{31}$$

where $\mathbf{d}_g, \mathbf{d}_l$ are target distances for global and local edges respectively, and $\lambda_g, \lambda_l$ are weighting factors.

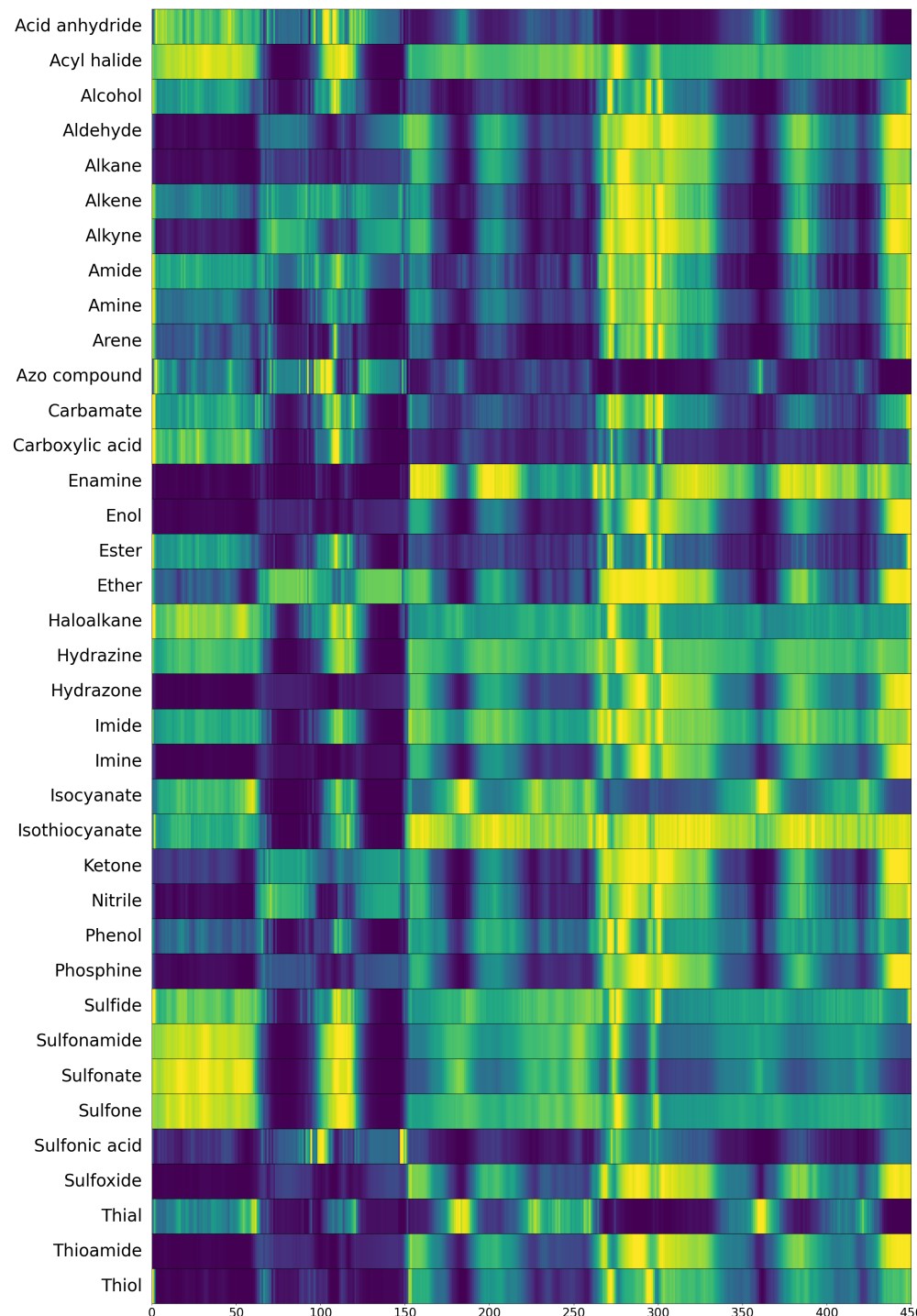

Figure 7: Illustration of the importance of spectral regions. The input spectrum is partitioned as follows: $[0, 150]$ corresponds to IR spectra, $[151, 300]$ to $^1$H-NMR spectra, and $[301, 450]$ to $^{13}$C-NMR spectra. Warmer colors indicate crucial (high-importance) information, while cooler colors represent redundant (low-importance) information.

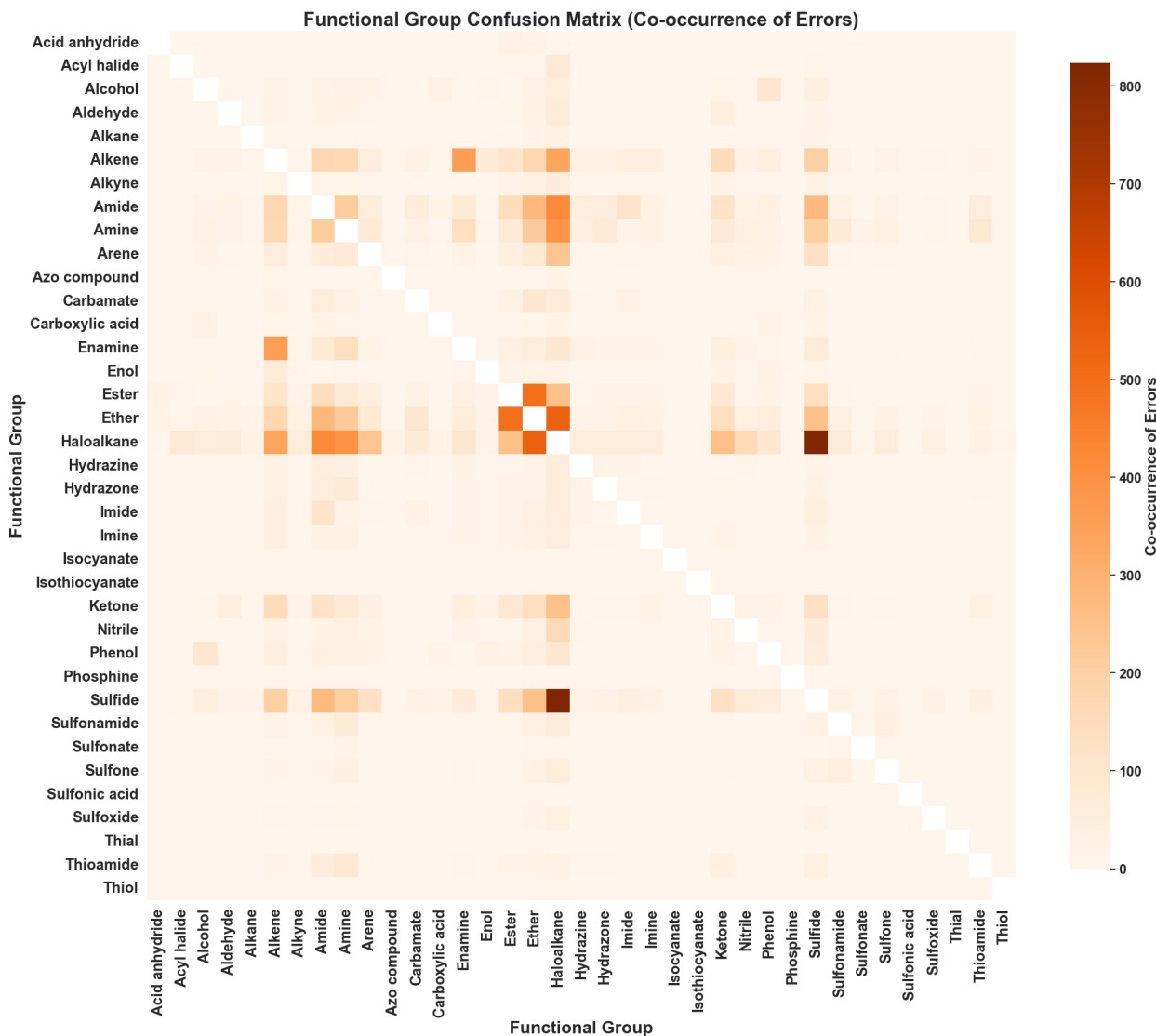

Figure 8: The confusion matrix between functional groups: the darker the color in the blocks, the higher the number of samples where the two functional groups were predicted incorrectly simultaneously.

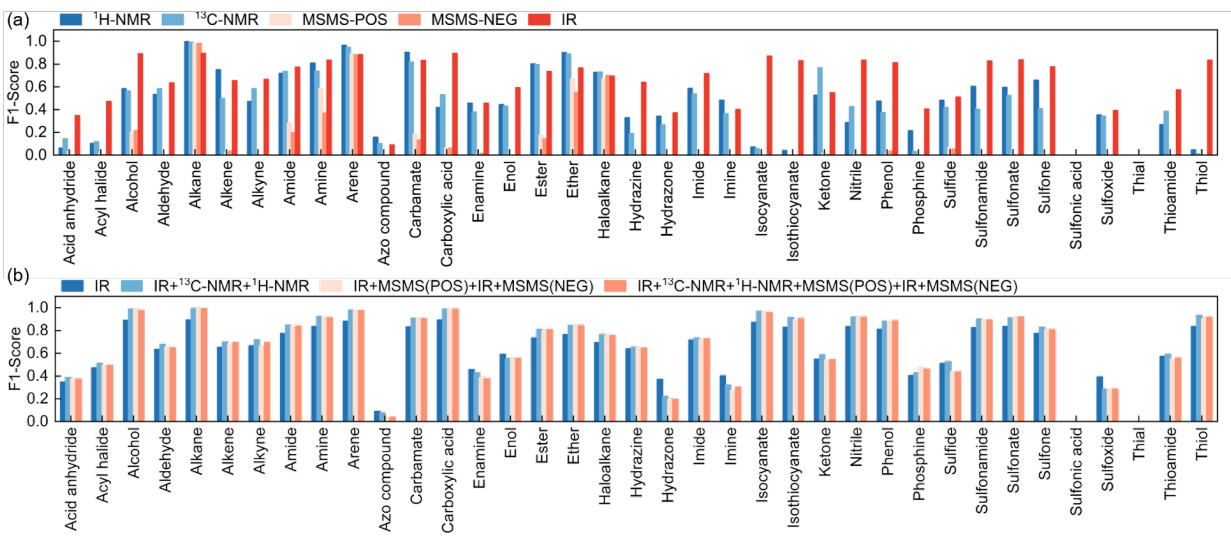

Figure 9: Performance of the model under unimodal and multimodal settings. (a) Results using a single spectrum as input (unimodal); (b) Results under multimodal fusion of spectra.

