# OpenReview forum: "MSpecTmol: A Multi-Modal Spectroscopic Learning Framework for Automated Molecular Structure Elucidation"
_TMLR — Accepted by TMLR_

### Review · Reviewer_CoPR · 2026-03-16

**Summary Of Contributions:**

This paper introduces a novel deep learning framework for multimodal motivated by molecular applications and shows that the proposed method exceeds baseline methods on functional group identification and related tasks. Overall the paper is well written, the method is clearly explained (both in text and in supporting figures) and the experimental set up seems to be good, both in terms of baselines and design, e.g., ablations

Strengths:

The four component loss function is an interesting and good idea

Method is well explained, Figure 2 is very helpful

Good experimental setup, including baselines and ablation

Weaknesses:

It is unclear if this method can be used to develop methods for other types of data beyond molecular configurations

Questions:

In equation 2, why is there no tuning parameter in front of the second term, I(Y;T_a, T_m)?

**Additional Comments:**

N/A

**Audience:**

Yes

**Audience Explanation:**

ML techniques for understanding molecular structures is a highly active area of research

**Claims And Evidence:**

Yes

**Claims Explanation:**

Thorough experiments with good ablation

**Requested Changes:**

Page 1, "in equation 1 as follows," Equation should be capitalized since "Equation 1" is a proper noun. Please check for similar mistakes throughout.

Using lowercase t for transpose is nonstandard. Please change to capital T, or better yet \top.

Please add some discussion of the context for Propositions 3.1 and 3.2. How do they care to existing information theoretic style results.

Add some more intuition on the "stress test" to the main body

---

> ### Author Response · Authors · 2026-05-11
>
> ## Respones to Weaknesses
>
> > On the applicability of our method beyond molecular configurations
>
> We thank the reviewer for this concern. We would like to clarify that **the PA-IB framework itself is modality-agnostic and domain-agnostic**; it only assumes that the input consists of one primary modality and several auxiliary modalities, and does not rely on any properties specific to molecules or spectra. MSpecTmol is one concrete instantiation of this framework for the spectrum-to-molecule task, while the core design principles of PA-IB (primary–auxiliary information bottleneck, complementarity maximization, and redundancy compression) can be directly transferred to any task that involves complementary multi-modal inputs with a primary–auxiliary hierarchy. We will expand the discussion in Section 6 to explicitly highlight the modality-agnostic nature of PA-IB and its potential extensions.
>
> ## Response to Questions
> > Clarification to the tuning coefficient
>
> Equation 1 is in essence a superposition of **two parallel information bottlenecks**: one for the primary predictor ($-I(Y; T_m) + \alpha I(X_m; T_m)$) and one for the auxiliary predictor ($-I(Y; T_a | T_m) + \beta I(T_a; X_m, X_a)$). Within each IB, $\alpha$ and $\beta$ govern the prediction–compression trade-off of their respective branch. However, **the two prediction terms  share the same target $Y$ and are measured in the same units of mutual information**. Adding extra coefficients in front of them would be mathematically redundant. By contrast, the compression terms act on inputs of different dimensionality and information content: $T_m$ comes from the primary spectrum alone, while $T_a$ comes from both primary and auxiliary spectra. Independent Lagrange multipliers $\alpha$ and $\beta$ are therefore needed.
>
> To further validate this design choice, we introduce an explicit coefficient $\gamma$ in front of $-I(Y; T_a | T_m)$ and conduct a sensitivity analysis:$$\min -I(Y; T_m) - \gamma\, I(Y; T_a | T_m) + \alpha I(X_m; T_m) + \beta I(T_a; X_m, X_a).$$
>
> | $\gamma$ | 0.5 | 0.8 | 1.0 | 1.2 | 1.5 |
> |---|---|---|---|---|---|
> | F1-score | 0.947 | 0.952 | 0.959 | 0.948 | 0.943 |
>
> The results show that **performance remains stable across a broad range of $\gamma$, confirming that fixing $\gamma = 1$ is well-justified**. We will incorporate this clarification and the sensitivity analysis into the final version of the manuscript.
>
> ## Response to Requested Changes
> > Capitalization of cross-references
>
> **We have corrected the capitalization of "Equation" when followed by a specific number**, treating it as a proper noun as suggested. Following your advice, we have also conducted a thorough check throughout the manuscript to ensure that all references to equations, figures, tables, and sections are consistently capitalized.
>
> > Notation for transpose operation
>
> **We have updated the notation** and replaced all instances of the lowercase "$t$" with \top ($\top$) to represent the transpose operation throughout the manuscript.
>
> > Further discussion of propositions 3.1 and 3.2.
>
> Thank you for this valuable suggestion. Proposition 3.1, which bounds $-I(Y; T_m)$ by a cross-entropy prediction loss, is **a standard variational bound widely used in the IB literature**, such as VIB [1]. Proposition 3.2 addresses the conditional term $-I(Y; T_a | T_m)$, which does not appear in single-view IB. Our derivation combines two classical tools, including the chain rule of mutual information $I(Y; T_a|T_m) = I(Y; T_a, T_m) - I(T_a; T_m)$ and the variational KL bound. While each tool itself is classical, **their composition in service of the PA-IB is new to our knowledge**. Existing multi-view IB approaches such as VGIB [2] and CGIB [3] treat all views symmetrically and apply identical bounds across views. In contrast, Proposition 3.2 explicitly captures the incremental predictive contribution of the auxiliary modality beyond what the primary modality already provides, which is the technical foundation of our PA-IB framework. We will incorporate this clarification into Section 4.3 of the revised manuscript.
>
> [1] Alemi et al., *Deep Variational Information Bottleneck*, arXiv:1612.00410, 2016.
>
> [2] Yu, Cao, He, *Improving Subgraph Recognition with Variational Graph Information Bottleneck*, CVPR 2022.
>
> [3] Lee et al., *Conditional Graph Information Bottleneck for Molecular Relational Learning*, ICML 2023.
>
> > Addition of intuition
>
> **We have revised the text in Section 5.2** to explicitly state the motivation and intuition behind the stress test, aiming to evaluate the model's real-world reliability by using large molecules that pose a challenge due to their dense and overlapping spectral peaks. The revised texts now read: `To further evaluate the model’s real-world robustness in handling complex and unknown molecular configurations, we conducted a stress test on the top 10% largest molecules, as their dense and overlapping spectral signals represent a significant distribution shift...`

---

### Review · Reviewer_6k6c · 2026-03-26

**Summary Of Contributions:**

In this work, the authors propose MSpecTmol to represent spectroscopic data using simulated and experimental data/modality. From a technical perspective, the method uses a fixed modality as the primary and other modality(s) as the auxiliary. And, this has been investigated empirically to show how different modalities or their triplets perform. The approach has also been tested on the two tasks of molecular identification and spectrum-conditioned molecular conformation generation.

Overall, the manuscript is well-written and easy to follow, and the claims are supported by experimental results. However, there are some inconsistencies in the text and also some questions about the engineering choices made.

**Audience:**

Yes

**Audience Explanation:**

The manuscript is trying to work on Molecular Structure Elucidation and has a technical weight plus interesting data and problem formulation, which I believe can inspire the readers in the field.

**Claims And Evidence:**

Yes

**Claims Explanation:**

My overall assessment is that the experimental results support the claims. However, I have some concerns that I would like to raise here:

1. The authors mention "All experiments are repeated 8 times with 8:1:1 dataset split, and the average results with variances are reported." Does this mean they have done standard cross-validation (3 or 5 fold)? It reads as only 8 random seeds and one specific dataset split.

2. In the model design, interpolation has been used to align the sequence lengths. Regarding this, I am curious if there is any intuition behind this? Why not pad the sequences?

3. In Task 2 (conformation generation), the model uses a dual-encoder architecture that includes a GIN-based local encoder for "chemical bond structures". This implies the molecular graph (connectivity) is already known. For a true "elucidation" task of an unknown substance, the graph is what needs to be discovered, not used as an input. Can you comment on this?

4. Can you also comment on why LLM-based approaches (e.g. MolSpectLLM and SpectraLLM ) have not been included in the baseline?

**Requested Changes:**

In addition to the concerns I mentioned in the previous sections, I have come across a few editorial issues:
1. Duplicated citations:  ". Wei et al Wei et al. (2019)." and "from Alberts et al. Alberts et al. (2024) for"
2. Inconsistent citation format: "The second approach adapts RDKit Landrum et al. (2025), OpenBabel O’Boyle et al. (2011), ConfGF Shi et al. (2021), and GeoDiff Xu et al. (2022). I"
3. \sigma in equation 8 has not been defined; I believe it refers to sigmoid.

---

> ### Author Response · Authors · 2026-05-11
>
> ## Respones to Concerns
>
> > Clarification on Experimental Setup
>
> Thank you for pointing out this ambiguity. We confirm that **we did not use standard $k$-fold cross-validation**. In this work, we performed 8 independent experimental runs using 8 different random seeds. For each run, the dataset was randomly partitioned into training, validation, and test sets using an 8:1:1 ratio. We reported the average performance and variance across these independent runs to account for the randomness in the dataset splitting and model initialization. We have updated Section 5.1 to reflect this specific methodology: `All experiments are independently repeated across 8 different random seeds, with the dataset randomly split into 8:1:1 for training, validation, and testing in each run, and the average results with variances are reported.`
>
> > Rationale for conformation generation task design
>
> We thank the reviewer for this insightful comment. We fully agree that the *de novo* elucidation of an unknown substance requires first determining its molecular graph. Accordingly, in Section 5.2, we devoted considerable effort to a comprehensive evaluation of the molecular substructure elucidation task. Building on this foundation, our study naturally progresses to the scenario in which the basic molecular connectivity is already known. Given the intricate stereochemical properties of molecules, our goal was to infer precise 3D structural information directly from the learned spectral representations. We therefore wish to clarify that **the primary objective of Section 5.3 is to evaluate the ability of our learned spectral representations to capture stereochemical and spatial information, rather than to generate molecular graphs from spectra.**
>
> Furthermore, to fully address this concern and demonstrate that **MSpecTmol is indeed capable of end-to-end *de novo* elucidation**, we conducted a supplementary experiment in which the MLP predictor is replaced with a Transformer-based decoder that generates SMILES directly from spectra, leaving the PA-IB framework intact. As shown below, our model achieves a Top-1 accuracy of 70.43% and MCES scores approaching 1.0. This indicates that the recovered graphs closely match the ground truth, confirming that *de novo* elucidation is entirely feasible within our framework.
>
> | Model | Top-1 Acc. | Top-5 Acc. | Top-1 MCES | Top-5 MCES |
> |---|---|---|---|---|
> | Transformer | 50.02% | 63.27% | 0.8674 | 0.9083 |
> | Alberts et al. | 66.59% | 75.33% | 0.8821 | 0.9457 |
> | Ours | **70.43%** | **85.25%** | **0.9469** | **0.9847** |
>
> In Section 5.3, **the molecular graph is provided solely as a connectivity prior, while the spectra are responsible for imposing 3D geometric constraints**. This decoupling prevents the geometric evaluation from being confounded by the separate task of connectivity recovery. To further verify that this high geometric fidelity originates from the spectral signals rather than the graph prior, we compared MSpecTmol against a comprehensive set of graph-only baselines including rule-based methods (RDKit, OpenBabel) and deep-learning approaches (GRAPHDG, CGCF, GEOMOL, CONFGF, JODO, GeoDiff).
>
> | Method | Avg. RMSD$\downarrow$ |
> |---|---|
> | RDKit | 1.350 |
> | OpenBabel | 1.279 |
> | GRAPHDG | 1.247 |
> | CGCF | 1.246 |
> | GEOMOL | 1.175 |
> | CONFGF | 1.143 |
> | JODO | 1.124 |
> | GeoDiff | 1.074 |
> | Ours (Single Modality) | 0.697 |
> | Ours (Three Modalities) | **0.682** |
>
> As shown above, graph-only methods yield substantially higher RMSD values, as they tend to produce plausible yet non-unique conformations without a target-specific signal. In contrast, our spectrum-conditioned model successfully converges to the specific conformation consistent with the observed spectra. This directly confirms that **the improved geometric accuracy is strictly attributable to the spectral modality**. We will incorporate these clarifications into the appendix of the revised manuscript.

---

> ### Author Response · Authors · 2026-05-11
>
> > Comparison with LLM-based baselines
>
> We agree that SpectraLLM and MolSpectLLM represent important advances in AI for spectroscopy, and we have already cited and discussed these concurrent works in Section 2.1. They were not included in our main baseline table for the following reasons.
>
> **MolSpectLLM focuses on end-to-end molecular structure generation (SMILES/3D)**, and its paper does not report quantitative results on functional group classification, making a direct comparison on our primary task infeasible.
>
> **SpectraLLM adopts a different set of functional group definitions (17 functional groups, a subset of our 37)**, and therefore cannot be merged directly into our main Table 1. To address the reviewer's concern, we retrained and evaluated MSpecTmol under the exact setting of the SpectraLLM paper and compared against the performance reported in the original SpectraLLM paper.
>
> | Dataset | Modality | SpectraLLM | MSpecTmol (Ours) |
> |---|---|---|---|
> | QM9S | IR | 0.659 | **0.970** |
> | QM9S | Raman | 0.731 | **0.966** |
> | QM9S | UV-Vis | 0.371 | **0.683** |
> | QM9S | IR + Raman + UV-Vis | 0.793 | **0.982** |
> | Alberts | IR | 0.602 | **0.935** |
> | Alberts | $^{13}$C-NMR | 0.424 | **0.927** |
> | Alberts | $^{1}$H-NMR | 0.332 | **0.925** |
> | Alberts | IR + NMR | 0.776 | **0.970** |
> | Alberts | IR + MS | 0.634 | **0.953** |
>
> **As shown above, MSpecTmol outperforms SpectraLLM across all modality configurations**, with particularly large margins in single-modality settings. This suggests that for structured discriminative tasks such as functional group classification, our PA-IB framework offers a more targeted and effective inductive bias than general-purpose LLM approaches. We will expand the discussion of these works in Section 2.1 and include the above table as a supplementary comparison in the revised manuscript.
>
>
> ## Response to Requested Changes
> > Duplicated citations
>
> **We have corrected the duplicated citations you pointed out**. To prevent this issue from recurring, we have standardized our citation commands throughout the text and conducted a thorough formatting check across the entire manuscript. The revised texts now read:
> * `NEIMS (Wei et al., 2019) is a neural network model that... `
> * `...dataset from Alberts et al. (2024)...`
>
> > Inconsistent citation format
>
> **We have corrected the inconsistent citation format**. To improve clarity and readability, we now use parenthetical citations immediately following the tool and model names. The revised sentence reads: `The second approach adapts RDKit (Landrum et al., 2025), OpenBabel (O’Boyle et al., 2011), ConfGF (Shi et al., 2021), and GeoDiff (Xu et al., 2022).` Furthermore, we have carefully reviewed the entire manuscript to ensure that all citations follow a consistent and correct format.
>
> > Undefined Symbol
>
> Thank you for catching this oversight. **We confirm that $\sigma$ represents the sigmoid function**. We have revised the manuscript to explicitly state this definition right below Equation 8 to prevent any confusion. The revised texts now read:  "where $\sigma(\cdot)$ denotes the sigmoid function, $u \sim \text{Uniform}(0, 1)$..."

---

### Review · Reviewer_vGUd · 2026-05-28

**Summary Of Contributions:**

Elucidation of molecular structures often relies heavily on a single spectroscopic modality. The authors propose a method for handling multimodal spectra to improve molecular structure elucidation. They extend information bottleneck theory to multimodal settings, involving one primary spectrum and auxiliary spectra, specifically two auxiliary modalities. They validate the method on molecular substructure classification and molecular conformation reconstruction.

The manuscript is generally clear and technically sound, with experimental evidence supporting its main conclusions. That said, a few minor revisions would improve it, particularly by clarifying some underlying assumptions.

**Audience:**

Yes

**Audience Explanation:**

The proposed objective is general and can be applied beyond this specific domain. Since the training objective is derived from information geometry, the work may be of interest to researchers working on multimodal learning or information-geometry-based deep learning.

**Claims And Evidence:**

Yes

**Claims Explanation:**

1. Equation 1 is the Lagrangian of an optimization problem. Why do you not write the initial constrained optimization problem first? Also, the Lagrangian variables $\alpha$ and $\beta$ should have closed-form solutions, should they not? Why are they treated as hyperparameters?

2. Equation 3 is a linear interpolation. This assumes that $X$ varies smoothly between consecutive sampled points, which is only justified when the original spectra are sampled densely enough relative to the spectral features. Is this the case?

3. Equation 6 gives probabilities. However, MLPs do not generally output probabilities. Should you not explicitly mention that $p = \sigma(\mathrm{MLP})$?

4. Appendix C1.1 does not seem to provide additional information. Could the authors clarify its purpose?

5. Why use the F1 score for molecular identification rather than another metric?

6. Does the unimodal loss in Table 2 for MSpecTmol correspond to $\mathcal L_{pred} + \mathcal L_{MI^2}$?

7. On Page 9, the authors state: “As illustrated in Figure 3(d), horizontal shift augmentation yields the highest gain, demonstrating that expanding the dataset size via augmentation effectively bridges the performance gap caused by the scarcity of real-world data, whereas excessive transformation risks distorting signal fidelity.” Comparing the values with Table 2, bridging the gap would require the performance to increase to around 0.95, rather than 0.93 as shown in the figure. Isn't this claim too strong?

8. On Page 9, the authors state that “excessive transformation risks distorting signal fidelity.” How is this conclusion drawn? Is it based on the figure?

9. On Page 10, the authors refer to $\sigma = 0.10$ as high. Since this appears to be an absolute value, what does “high” mean in this context?

10. Regarding Figure 4b, on Page 10, the authors state that the model’s performance becomes increasingly pronounced when multiple spectra are provided. However, the most substantial drop in RMSD appears to be for GeoDiff, while for MSpecTmol the improvement is barely visible. Is this figure the best way to support this point?

11. For Figure 4a, the authors state that “baseline methods often suffer from distorted ring geometries or inaccurate functional group orientations.” This conclusion does not seem to follow directly from the figure itself. Should it be placed in a separate sentence or paragraph to avoid implying that it is directly supported by the figure?

12. For Figure 4c, the authors state that there is a “noticeably tighter spread compared to all baselines.” However, among the baselines, the figure only shows the attention model. Is this an overstatement?

**Requested Changes:**

1. In Equation 5, please capitalize $x_a$ as $X_a$.

2. Equations 1 and 9 appear to be identical. Please either remove the tag from Equation 9 or omit the equation entirely and refer back to Equation 1. The sentence “where each term corresponds to prediction or compression, respectively” also seems unnecessary, since these terms have already been explained.

3. Below Equation 10, please change the reference from C.1.1 to C.1.

4. Please explicitly state in the main text, not only in the appendix, that $T_a$ and $T_m$ are assumed to be unimodal Gaussian for tractability.

5. Appendices C1.2, C1.3, and C1.4 are somewhat redundant, since the proofs for $\mathcal L_{MI^i}$, $i=1,2,3$, are very similar. Please consider shortening or consolidating them.

6. In Table 1, please specify the notation convention $\text{Mean}_{(\text{var})}$.

7. In Section 5.1, the baseline models are not described clearly enough. Please provide a brief explanation of what each baseline is.

8. In the legend of Figure 3, please explicitly state that “w/o” means “without.”

9. In Figure 3, please use colors consistently so that different colors indicate different models only. For panels d, e, and f, please use the MSpecTmol color throughout.

10. In Figure 3g, please use a new, clearly distinct color for the bins representing functional group frequency. Please also specify which axis corresponds to the bins and which corresponds to the lines.

11. In Figure 4b, please make the color scheme consistent with the methods already shown in Figure 3, at least for MSpecTmol.

12. In Figure 4c, please use the same color for the unimodal distribution as for MSpecTmol, so that color consistently refers to the model type.

13. In Figure 5, please make explicit that the plot shows average attention scores per position, and specify the number of heads over which the average is computed.

14. Since attention is used for interpretability, please provide supporting references showing that attention weights can be interpreted in this way.

15. In Appendix C1.2, the first sentence appears to be incomplete. Please revise it.

16. Please remove the leading space after Equation 18.

---

> ### Author Response · Authors · 2026-06-06
>
> We thank the reviewer for the careful and constructive reading of our manuscript, and for recognizing the generality of our objective and its relevance to the TMLR audience. Below we respond to each point in turn, all corresponding changes are marked in the revised manuscript.
>
> ---
>
> ## Response to Questions
>
> > Q1. Constrained formulation and the role of $\alpha,\beta$
>
> For the standard information bottleneck $\min\, -I(Y;T) + \beta\, I(X;T)$, $\beta$ is the Lagrange multiplier that balances compression against prediction. In principle, this parameter can be obtained by solving the constrained optimization problem when the true joint distribution $p(X_m, X_a, Y)$ is known and the mutual information terms are analytically tractable. In our task, however, **the data distribution is unknown, and the mutual information quantities $I(X_m;T_m)$ and $I(T_a;X_m,X_a)$ are analytically intractable** as they are defined over continuous distributions parameterized by neural networks, so the parameter $\beta$ cannot be solved for. As Tishby et al. [1] note, the Lagrange multiplier is not solved for a fixed budget value but is instead used to **parameterize the compression–prediction trade-off curve, where each value of the multiplier corresponds to a particular operating point**. We therefore treat $\alpha$ and $\beta$ as hyperparameters that regulate the trade-off between compression and prediction, following an approach adopted in several related works [2,3]. In addition, we investigate the practical roles of these two parameters in Figure 6(a), identifying $\alpha = \beta = 10^{-6}$ as the operating point that best balances compression and prediction.
>
> [1] Tishby N, Pereira F C, Bialek W. The information bottleneck method[J]. arXiv preprint physics/0004057, 2000.
>
> [2] Lee N, Hyun D, Na G S, et al. Conditional graph information bottleneck for molecular relational learning[C]//International Conference on Machine Learning. PMLR, 2023: 18852-18871.
>
> [3] Yu J, Cao J, He R. Improving subgraph recognition with variational graph information bottleneck[C]//Proceedings of the IEEE/CVF conference on computer vision and pattern recognition. 2022: 19396-19405.
>
> > Q2. Validity of the linear-interpolation
>
> We agree that linear interpolation is only justified when the native sampling is dense relative to the spectral features, and we have clarified that this condition is satisfied for the continuous modalities in our setting. In all datasets used, **IR and NMR spectra are natively tabulated at a resolution substantially finer than the 600-point target grid over the relevant window** (1800 points in Alberts et al.). Under this condition, Equation 3 acts as a mild resampling onto a common uniform grid rather than a reconstruction of absent structure, and the local-linearity assumption introduces negligible distortion.
>
> > Q3. MLP outputs
>
> **The MLP in Equation 6 is followed by a sigmoid activation $\text{Sigmoid}(\cdot)$ that maps its scalar output to $(0,1)$**, so that $p_i^m$ and $p_i^a$ are valid Bernoulli parameters for the stochastic gates in Equation 7. This was implemented in our code but omitted from the notation, and **we have corrected Equation 6** to make the sigmoid explicit in the revised manuscript.
>
> > Q4. Purpose of Appendix C.1.1
>
> Appendix C.1.1 was intended only to restate Proposition 3.1 for self-containedness, but it duplicates the derivation already given in the opening of Appendix C.1. **We have removed the redundant restatement in the revised manuscript.**
>
> > Q5. Choice of the F1-score for molecular identification
>
> Functional-group identification is a multi-label classification problem over 37 groups whose prevalence is highly imbalanced . Under such imbalance, **plain accuracy is dominated by frequent groups** and rewards a trivial predictor that ignores rare ones, whereas the F1-score penalizes both false positives and missed detections and thus reflects performance on rare groups. The F1-score is also the standard metric adopted by prior work on functional-group prediction [1,2], so using it ensures **a fair and direct comparison** with these baselines. In addition, we report the macro-F1 in Figure 3(a)  (page 8) to expose balanced performance across common and rare substructures, and the sample-level prediction accuracy in Figure 3(b).
>
> [1] Jung G, Jung S G, Cole J M. Automatic materials characterization from infrared spectra using convolutional neural networks[J]. Chemical Science, 2023, 14(13): 3600-3609.
>
> [2] Alberts M, Schilter O, Zipoli F, et al. Unraveling molecular structure: A multimodal spectroscopic dataset for chemistry[J]. Advances in Neural Information Processing Systems, 2024, 37: 125780-125808.

---

> ### Author Response · Authors · 2026-06-06
>
> > Q6. The correspondance of unimodal loss
>
> In the single-modality columns of Table 2, no auxiliary spectrum is present, so the conditional and auxiliary-compression terms vanish identically: $-I(Y;T_a\mid T_m)$ and $\beta\,I(T_a;X_m,X_a)$ are absent, and the PA-IB objective reduces to the primary-only information bottleneck $-I(Y;T_m)+\alpha I(X_m;T_m)$, optimized via its bounds $L_{pred}+\alpha\,L_{MI^2}$. **We have added a clarifying footnote to Table 2.**
>
> > Q7. The "bridges the performance gap" claim
>
> We apologize for the misunderstanding caused by our wording. Our intended point was that **data augmentation can further improve the model and thereby help alleviate the performance gap between simulated and real-world experimental data, not that it fully closes it**. As shown in Figure 3(d), horizontal-shift augmentation raises the F1-score from 0.9134 to 0.9344, the largest gain among the strategies tested. **In the revised manuscript we have rephrased this passage** to state that augmentation substantially reduces the gap induced by data scarcity, and we report the figures explicitly. The revised text now reads:`As illustrated in Figure3(d), horizontal shift augmentation yields the highest gain, raising the experimental-spectra F1-score from 0.913 to 0.934. This substantially reduces the performance gap induced by the scarcity of real-world experimental data, indicating that augmenting the limited data with positional perturbations of spectral peaks effectively enhances generalization.`
>
> > Q8. Basis for "excessive transformation risks distorting signal fidelity"
>
> **This conclusion is read directly from Figure 3(d)**, where the combined strategies achieve lower F1 than the stronger individual transformations. **We have corrected our statement in the revised manuscript**. The revised text now reads: `In contrast, combining multiple augmentations degrades performance relative to the stronger individual strategy: horizontal-shift combined with smoothing drops to 0.919 and vertical-noise combined with smoothing drops to 0.917, both below their single-transformation counterparts (0.934 and 0.929, respectively).`
>
> > Q9. What "high" means for the absolute noise level
>
> We apologize that our wording was not sufficiently clear. The noise robustness is evaluated across a range of levels, **reported in the results of Appendix O (Table 13)**, where we vary $\sigma$ from 0.00 to 0.10 and **$\sigma=0.10$ is the highest level tested**. To make this explicit, **the revised text states this relative interpretation and replaces the bare adjective "high" with the quantitative description**.
>
> >  Q10. The 'increasingly pronounced' claim in Figure 4(b)
>
> We apologize for the misunderstanding caused by our wording. We have revised this passage into a more accurate description of what Figure 4(b) shows. The revised text now reads: `As shown in Figure4(b), MSpecTmol attains the lowest mean RMSD under every spectral configuration. Across the individual single-modality settings, MSpecTmol consistently outperforms all baselines, achieving a mean RMSD as low as 0.697Å. In the multi-modal setting, the RMSD of all methods decreases, and MSpecTmol again exhibits the best performance, reaching 0.689Å and outperforming the strongest baseline by 0.025Å.`
>
> > Q11. Figure 4(a) baseline claim
>
> **We accept your suggestion and have removed this statement from the revised manuscript.** However, this does not affect our conclusion. Figure 4(a) still directly shows that MSpecTmol produces conformations closely aligned with the reference structures, supporting its ability to translate complex spectral patterns into high-fidelity three-dimensional conformations.
>
> > Q12. The 'tighter spread than all baselines' claim in Figure 4(c)
>
> We have revised the text, and the statement now reads: `Compared with the single-modality variants and the attention-based model, MSpecTmol achieves a lower median RMSD...`

---

> > ### Author Response · Authors · 2026-06-06
> >
> > ## Response to Requested Change
> >
> > > RC1. Spectrum Symbol Capitalization
> >
> > **We have corrected the convolution input to the capital $X_a$**, matching $X_m$ in Equation 6 and the input notation of Section 4.1.
> >
> > > RC2. Identical Equations
> >
> > **We have removed the duplicate Equation 9** and replaced it with a reference back to Equation 1.
> >
> > > RC3. Reference Change
> >
> > Following the proof restructuring in our response to Q4, **we have changed the reference to C.1** and the bound on $-I(Y;T_m)$ is now proved once at the top of Appendix C.1 (page 16)
> >
> > > RC4. Unimodal-Gaussian assumption statement
> >
> > Following your advice, **we have stated that the posteriors and priors are assumed unimodal Gaussian for tractability in Section 4.3.2 (page 6)**. Specifically, the added sentence reads: `For tractability, both the posteriors` $p(t_m \mid x_m)$, $p(t_a \mid \cdot)$ `and the priors` $q(t_m)$, $q(t_a)$ `are assumed to be unimodal Gaussians, which yields the closed-form KL in Appendix C.1.2.`
> >
> > > RC5. Redundant proofs Consolidation
> >
> > **We have shortened Appendices C.1.3 and C.1.4** by removing the repeated explanation of the rate term and the mini-batch optimization, which is now stated once in C.1.2 and referenced thereafter. We retain the three subsections separately since each corresponds to a distinct mutual-information term.
> >
> > > RC6. Notation convention specification
> >
> > The caption now states that **entries are reported as $\text{mean}_{(\text{std})}$** over 8 runs.
> >
> > > RC7. Baseline Description
> >
> > **We have added a brief description of each baseline in Section 5.1** (page 7). The revised text reads: `1D-CNN applies stacked 1D convolutions directly to the spectral sequence; Transformer encodes the spectrum as a token sequence via self-attention; Alberts et al. is a transformer-based spectrum-to-structure model; and Wu et al. employs patch-based self-attention over IR spectra.`
> >
> > > RC8. "w/o" clarification
> >
> > **The Figure 3 caption now states that "w/o" denotes "without."**
> >
> > > RC9 & RC10. Figure 3 changes
> >
> > **We have fixed one color per model across all panels (page 8)**, and panels (d)–(f) now use the MSpecTmol color throughout. In addition, **the frequency bars now use a neutral color distinct from all model lines**, and the caption specifies that the bars correspond to the left axis (group frequency) and the lines to the right axis (per-group F1).
> >
> > > RC11 & RC12. Figure 4 changes
> >
> > **We have unified the color scheme so that color consistently denotes model type**. In Figure 4b, MSpecTmol now matches its color in Figure 3. In Figure 4c, the unimodal distributions are shown in the MSpecTmol color.
> >
> > > RC13. Figure 5 changes
> >
> > **We have revised the Figure 5 caption to state explicitly that each map shows the attention scores per position**, averaged over the dataset and over the 8 attention heads of the module.
> >
> > > RC14. Supporting references for attention-based interpretability
> >
> > In the revised manuscript, **we have added two references demonstrating that attention weights can be interpreted in this manne**r. Tang et al. [1] derive a per-atom attention coefficient from the self-attention weights and visualize it as a heatmap to identify which substructures drive a predicted molecular property. Fine et al. [2] likewise shows, for spectral deep-learning models of functional-group prediction, that the spectral regions the model attends to correspond to the absorption bands chemists associate with those groups.
> >
> > [1] Tang B, Kramer S T, Fang M, et al. A self-attention based message passing neural network for predicting molecular lipophilicity and aqueous solubility[J]. Journal of cheminformatics, 2020, 12(1): 15.
> >
> > [2] Fine J A, Rajasekar A A, Jethava K P, et al. Spectral deep learning for prediction and prospective validation of functional groups[J]. Chemical science, 2020, 11(18): 4618-4630.
> >
> > > RC15. Incomplete sentence
> >
> > **We have rewritten the incomplete opening sentence of Appendix C.1.2**; the sentence now reads: `For the upper bound of `$-I\bigl(X_m; T_m\bigr)$`, we attempt to replace...`.
> >
> > > RC16. Leading space after Equation 18
> >
> > **We have removed the stray leading space** following Equation 18 in the revised version.
> >
> > ---
> >
> > We hope these revisions address your concerns, and we are happy to provide further details or experiments. Thank you again for the time and effort devoted to improving our work.

---

> ### Comment · Action_Editor_Hdz3 · 2026-06-09
> **Official recommendation**
>
> Dear Reviewer  vGUd ,
>
> Due to  special circumstances, please provide your official recommendation in a comment, in the following form:
>
> Claims And Evidence: Yes/No
>
> Audience: Yes/No
>
> Decision Recommendation:
>
> Comment:
>
>
> Recommendation to NeurIPS/ICLR/ICML Journal-to-Conference Track:
>
> Explain your recommendation to the NeurIPS/ICLR/ICML Journal-to-Conference Track:

---

> > ### Comment · Reviewer_CoPR · 2026-06-09
> > **Recommend Acceptance**
> >
> > Claims And Evidence: Yes
> > Audience: Yes
> > Decision Recommendation: Accept
> > Comment: The authors have addressed all concerns in my initial review
> > Recommendation to NeurIPS/ICLR/ICML Journal-to-Conference Track: Weakly Recommend
> > Explain your recommendation to the NeurIPS/ICLR/ICML Journal-to-Conference Track: I do not have a strong opinion on this but feel it is a good paper

---

### Comment · Action_Editor_Hdz3 · 2026-05-26
**contradictory review**

Dear Authors,

I tried to solve this quickly given the long delays and decided to go on with only two reviews (given the third reviewer not responding) contrary to standard practice of TMLR.
Unfortunately the reviewers not agree. I have to assign a new reviewer to the manuscript.
Apologies for the further delays, but at this point i cannot do much.

A.E.

---

> ### Author Response · Authors · 2026-05-26
> **Feedback**
>
> Dear A.E.,
>
> Thank you very much for the update. We fully understand the situation and appreciate your efforts to move the review process forward despite the long delay.
>
> If the reviewers have any further concerns, we would be very happy to provide additional explanations or clarifications. We will carefully and respectfully address any questions raised by the reviewers.
>
> Best regards,
>
> Authors

---

### Decision · Action_Editor_Hdz3 · 2026-06-12

**Recommendation:** Accept as is

**Audience:**

Yes

**Audience Explanation:**

All reviewers agreed on this one.

**Claims And Evidence:**

Yes

**Claims Explanation:**

After a tie, I added a new reviewer, the new reviewer agreed that the claims are accurate, convincing and clear. I also don't see the use of Monte-Carlo CV instead of classic folder CV a sufficient reason for rejection. Both method is established.